# Learning Hyper Label Model for Programmatic Weak Supervision

**Renzhi Wu**[1], **Shen-En Chen**[1], **Jieyu Zhang**[2], **Xu Chu**[1]
[1]Georgia Tech [2]University of Washington
{renzhiwu@,achen353@,xu.chu@cc.}gatech.edu
jieyuz2@cs.washington.edu

## Abstract

To reduce the human annotation efforts, the programmatic weak supervision (PWS) paradigm abstracts weak supervision sources as labeling functions (LFs) and involves a label model to aggregate the output of multiple LFs to produce training labels. Most existing label models require a parameter learning step for each dataset. In this work, we present a hyper label model that (once learned) infers the ground-truth labels for each dataset in a single forward pass without dataset-specific parameter learning. The hyper label model approximates an optimal analytical (yet computationally intractable) solution of the ground-truth labels. We train the model on synthetic data generated in the way that ensures the model approximates the analytical optimal solution, and build the model upon Graph Neural Network (GNN) to ensure the model prediction being invariant (or equivariant) to the permutation of LFs (or data points). On 14 real-world datasets, our hyper label model outperforms the best existing methods in both accuracy (by 1.4 points on average) and efficiency (by six times on average). Our code is available at https://github.com/wurenzhi/hyper_label_model

## 1 Introduction

The lack of labeled training data is a major challenge impeding the practical application of machine learning (especially deep learning) techniques. Therefore, practitioners have been increasingly turned to *weak supervision* in which large amounts of cheaply generated noisy labels are used. There are many forms of weak supervision sources, e.g. external knowledge bases (Mintz et al., 2009), existing pre-trained models (Das et al., 2020; Wu et al., 2022b), and heuristics/rules (Shin et al., 2015). To unify different sources, the *programmatic weak supervision (PWS)* paradigm (Ratner et al., 2016; 2017; Zhang et al., 2022) was proposed. In PWS, the user expresses each available weak supervision signal from different sources with a labeling function (LF), a small program that takes in a data point and outputs a noisy label. After that, each LF is applied to unlabeled data of arbitrary size to obtain a noisy label vector; then, a label aggregation model (also referred as label model in literature) is used to aggregate all noisy label vectors to infer the unknown ground-truth labels. The inferred labels can then be used to train any downstream end models. The PWS paradigm has been successful in various tasks (Wu et al., 2018; Fries et al., 2019; Lison et al., 2020; Wu et al., 2021; 2020; Li et al., 2021) and industry scenarios (Mathew et al., 2021; Bach et al., 2019; Dunnmon et al., 2020).

The core challenge in PWS is how to aggregate all noisy label vectors to infer the ground-truth labels. Let label matrix $X$ denote the noisy labels where each column $X[:,j]$ denotes the noisy label vector from the $j^{th}$ LF and each row $X[i,:]$ denotes the weak labels of the $i^{th}$ data point; Let $\boldsymbol{y}$ denote the ground-truth label vector. Most existing label models assume an underlying distribution $p(\boldsymbol{y}[i]|X[i,:];\theta)$ (Zhang et al., 2022) where $\boldsymbol{y}[i]$ is the label for the data point and $\theta$ is the parameter of the distribution. The parameter $\theta$ is first learned on the weak labels $X = (X[1,:], X[2,:], \dots)$ in an unsupervised and typically iterative way, and then inference is made using $p(\boldsymbol{y}[i]|X[i,:];\theta)$. In this approach, the parameter $\theta$ is dataset-specific and has to be learned for every different $X$ (dataset).

In contrast to existing solutions, we propose a hyper label model with the goal of reducing assumptions and parameter learning process. Specifically, we aim to develop a hyper model that enjoys

two desiderata: (1) it works with "minimal" assumption, *i.e.*, we only assume the majority of LFs is better-then-random while does not require the knowledge or assume any particular forms of underlying distribution $p(\boldsymbol{y}[i]|X[i,:];\theta)$; (2) once the hyper model is learned, it can be used to infer $\boldsymbol{y}$ for any new $X$ without additional dataset-specific parameter learning process. To shed light on this direction, we first show, in theory, that without assuming underlying distribution, there is an optimal and analytical (therefore no parameter learning) way to estimate of $\boldsymbol{y}$ based on $X$, *i.e.*, $\boldsymbol{y}^* = h^*(X)$. However, such $h^*$ is intractable to compute since it involves averaging over a set whose size is exponentially-increasing *w.r.t.* the size of $X$. Therefore, we propose to leverage the power of deep learning to approximate this solution, *i.e.*, we seek for an alternative function $h$ parametrized by some neural networks, and once learned, it can estimate the label vector for new dataset without ad hoc dataset-specific learning process. Thus, we call the learned model *hyper label model*.

Materializing this idea involves two key questions: (1) How to generate training data? (2) How to design the model architecture? To generate training data, the straightforward solution is to use the analytical method to generate many pairs of $(X, \boldsymbol{y}^*)$ where $\boldsymbol{y}^* = h^*(X)$. However, computing $\boldsymbol{y}^*$ with $h^*(X)$ is of exponential complexity. We notice that for each $X$, $h^*(X)$ is an average of the label vectors from a certain set. Taking advantage of this, we are able to avoid directly generating $\boldsymbol{y}^*$ that is of exponential complexity and design a way of generating an equivalent set of training data such that the trained model approximates $h^*(X)$.

The model architecture has two requirements. First, it should be able to accept input matrix $X$ of arbitrary size as the size of the matrix $X$ can be different across datasets. Second, the output of the model (e.g. the predicted label vector) should be invariant to the permutations of columns in $X$ as the order of the LFs should not impact the final predicted labels; The output of the model should be equivariant to the permutation of rows in $X$ as when switching the order of the data points in $X$ the predicted labels should be switched accordingly. We noticed that a Graph Neural Network (GNN) is able to accept an input graph of arbitrary size and is permutation equivariant to the nodes on the graph (and can also be made to be permutation invariant by taking the average of the nodes). Therefore, we propose to represent the input matrix $X$ as a graph and then design a GNN to satisfy the two requirements.

**Contributions.** We make the following contributions:

(1) We for the first time present an analytical method for label aggregation which is optimal in the sense that it minimizes a certain form of the averaged prediction error, though directly using the analytical method is of exponential complexity.

(2) We train a model to learn the analytical method. The trained model is a *hyper label model* that can be used to infer the ground-truth labels for unseen datasets in a single forward pass without needing any dataset-specific parameter learning.

(3) We design a synthetic training data generation method and show that the hyper label model trained on the synthetically generated data learns to be the analytical method.

(4) We design an effective model architecture based on GNN so that the hyper label model is applicable to arbitrary number of LF label vectors of arbitrary size and is invariant/equivariant to the permutation of LF label vectors/data points.

(5) We show that our hyper label model outperforms the best existing methods over 14 real-world weak supervision datasets in both accuracy (by 1.4 points on average) and efficiency (by a speedup of six times on average) for both unsupervised and semi-supervised label aggregation.

## 2 RELATED WORK

All existing methods (except majority vote) first learn some parameter $\theta$ ad hoc for each new dataset and inference is then performed based on the learned parameter $\theta$. The existing methods differentiate from each other in how to formulate the parameter $\theta$ and how to learn the parameter (Zhang et al., 2022). For example, most methods assume an underlying distribution $p(\boldsymbol{y}[i]|X[i,:];\theta)$ (Ratner et al., 2016; 2019; Fu et al., 2020; Wu et al., 2022a; Yu et al., 2022) and focus on how to represent the distribution and how to learn the parameter $\theta$ of the distribution. Another example is that some approaches treat the accuracy of the LFs as parameters then use iterative methods to learn the accuracy parameters of the LFs (Arachie & Huang, 2021a;b; Dawid & Skene, 1979) for each

dataset. Different from all existing methods, our hyper label model directly performs inference on new datasets without needing an ad hoc dataset-specific parameter learning process.

In principle, $X$ could be any matrix in $\{+1, -1, 0\}^{n \times m}$ and $\boldsymbol{y}$ can be any vector in $\{+1, -1\}^n$. For arbitrary $X$ and $\boldsymbol{y}$, there is no way to infer $\boldsymbol{y}$ from $X$ with a better performance than random guess. Therefore, all label models implicitly or explicitly make some assumptions about the quality of the LFs. For example, assuming the accuracy of each LF is in certain range (Ratner et al., 2016) or the accuracy of LFs are known or can be estimated in a certain way (Arachie & Huang, 2021a;b; Dawid & Skene, 1979). On top of this, most existing methods also make additional assumptions about modeling. Specifically, most existing methods assumes a distribution $p(\boldsymbol{y}[i]|X[i,:]; \theta)$, then further assumes the distribution taking a certain form (e.g. probabilistic graphical models (PGM) (Ratner et al., 2016; Fu et al., 2020; Yu et al., 2022)). Our method only assumes the majority of the LFs is better than random guess, which is "minimum" comparing to existing methods.

While our work focuses on PWS, there are other methods to reduce annotation cost. One important line of work is self-supervised learning where feature representations are learned from self-defined pseudolabels and can then be used for downstream tasks (Jaiswal et al., 2020; Misra & Maaten, 2020; Liu et al., 2021). Another popular approach is active learning that interactively selects the most informative data points to annotate (Settles, 2012).

## 3 PROBLEM SETUP

Given a binary classification task, let $n$ and $m$ denote the number of data points and the number of LFs respectively. Let $X \in \{+1, -1, 0\}^{n \times m}$ denote a label matrix where $X[i,j] \in \{+1, -1, 0\}$ denotes the weak label of the $i^{th}$ data point provided by the $j^{th}$ LF. The values $+1$ and $-1$ denote the positive and negative classes respectively and $0$ denotes abstention, *i.e.* an LF does not have enough information to label a data point as either positive or negative (Ratner et al., 2016). The goal of a label model is to infer the unknown ground-truth label vector $\boldsymbol{y} \in \{+1, -1\}^n$ using $X$, which typically requires a learning process for each individual dataset, *i.e.*, $X$.

**The Better-than-random Assumption.** As discussed in Section 2, in weak supervision literature, there are often assumptions on the quality of LFs so that one can make a meaningful estimation of $\boldsymbol{y}$ using $X$. Different methods make different assumptions (Ratner et al., 2016; Fu et al., 2020; Ratner et al., 2019; Arachie & Huang, 2021a). In this work, we assume that for each class, the majority of LFs are better than random. This assumption is realistic since the LFs are typically made by human and humans might occasionally make mistakes when developing individual LFs, resulting in a small portion of worse-than-random LFs. Formally, this assumption can be expressed as:

$$\sum_{j=0}^{m-1} g(X, \boldsymbol{y}, j, +1) > \frac{m}{2} \text{ and } \sum_{j=0}^{m-1} g(X, \boldsymbol{y}, j, -1) > \frac{m}{2}, \tag{1}$$

where $g(X, \boldsymbol{y}, j, c)$ denotes whether the $j^{th}$ LF is better-than-random for class $c$:

$$g(X, \boldsymbol{y}, j, c) = \begin{cases} 1, \text{if } \sum_{i=0}^{n-1} \mathbf{1}_{X[i,j]=c \ \& \ \boldsymbol{y}[i]=c} > \sum_{i=0}^{n-1} \mathbf{1}_{X[i,j]=-c \ \& \ \boldsymbol{y}[i]=c} \\ 0, \text{otherwise}; \end{cases} \tag{2}$$

We define $\sigma(X, \boldsymbol{y}) = 1$ when Equation 1 is satisfied and $\sigma(X, \boldsymbol{y}) = 0$ otherwise. We say a pair $(X, \boldsymbol{y})$ is *valid* (or a vector $\boldsymbol{y}$ is *valid* for a given $X$) when $\sigma(X, \boldsymbol{y}) = 1$. Intuitively, $\sigma$ constrains the space of the predicted label vector $\hat{\boldsymbol{y}}$ and we would only predict one of label vectors with $\sigma(X, \hat{\boldsymbol{y}}) = 1$ for a label matrix $X$. Note the method we will propose is not tied to our better-than-random assumption, and it also works with any other assumptions to define $\sigma$.

**Our Goal.** Most existing methods aim to learn an individual label model for each dataset. In contrast, our goal is to learn a hyper label model under our better-than-random assumption. The learned hyper model can be applied to any unseen dataset and produces a prediction of the label vector $\boldsymbol{y}$ in a single forward pass without any form of dataset-specific parameter learning. Specifically, exisiting methods typically model the distribution $p(\boldsymbol{y}[i]|X[i,:]; \theta)$, where $X[i,:]$ is an individual data point from a dataset, *i.e.* one row in the label matrix $X$ (that represents all data points) and $\boldsymbol{y}[i]$ is

the corresponding label; The parameter $\theta$ should be learned for every new dataset before performing inference on each data point using the distribution $p(\boldsymbol{y}[i]|X[i,:];\theta)$. Instead, we aim to learn a hyper distribution $p(\boldsymbol{y}|X,\Theta)$ over all possible datasets with a hyper label model. Once the hyper label model has learned $\Theta$, for any new dataset $X_{\text{new}}$, it could directly produce prediction using the distribution $p(\boldsymbol{y}|X_{\text{new}},\Theta)$ without needing to learn a dataset-specific parameter $\theta$.

## 4 AN ANALYTICAL OPTIMAL SOLUTION

We first show there is an optimal and analytical (therefore no dataset-specific parameter learning) method to estimate $\boldsymbol{y}$ based on $X$, *i.e.*, $\boldsymbol{y}^* = h^*(X)$. This makes a hyper label model possible.

For each label matrix $X$, let $\mathcal{U}_{\boldsymbol{y}}(X) = \{\boldsymbol{y}|\sigma(X,\boldsymbol{y}) = 1\}$ denote the set of valid candidate $\boldsymbol{y}$s for $X$. The expected error of an estimator $h(X)$ of the $\boldsymbol{y}$ on each $X$ is:

$$\epsilon(X,h) = \sum_{\boldsymbol{y}\in\mathcal{U}_{\boldsymbol{y}}(X)} p(\boldsymbol{y}|X)||\boldsymbol{y} - h(X)|| \tag{3}$$

where $p(\boldsymbol{y}|X)$ is a distribution of $\boldsymbol{y}$ defined on set $\mathcal{U}_{\boldsymbol{y}}(X)$ and $||\cdot||$ denotes L2 loss (*i.e.* squared error). $p(\boldsymbol{y}|X)$ is unknown and can be different in different real-world applications (datasets). Without additional information apart from $X$, there is no way to determine the preference of some valid choices of $\boldsymbol{y}$ over other valid choices of $\boldsymbol{y}$, so the uniform distribution (*i.e.* $p'(\boldsymbol{y}|X) = \frac{1}{|\mathcal{U}_{\boldsymbol{y}}(X)|}$) is intuitively a good approximation for the unknown $p(\boldsymbol{y}|X)$. In fact, using the uniform distribution has optimalities in both the worst case and the average case. To maintain the flow of the paper, we defer the formal definition and proof of the optimalities of using the uniform distribution to Appendix A. Replacing $p(\boldsymbol{y}|X)$ by the uniform distribution, Equation 3 becomes:

$$\epsilon'(X,h) = \frac{1}{|\mathcal{U}_{\boldsymbol{y}}(X)|} \sum_{\boldsymbol{y}\in\mathcal{U}_{\boldsymbol{y}}(X)} ||\boldsymbol{y} - h(X)|| \tag{4}$$

$\epsilon'(X,h)$ can be interpreted as the average error of all possible outcomes. An estimator $h$ can be said to be the optimal if it minimizes the error $\epsilon'(X,h), \forall X$.

**Theorem 1.** $\forall X, h^*(X) = \frac{1}{|\mathcal{U}_{\boldsymbol{y}}(X)|} \sum_{\boldsymbol{y}\in\mathcal{U}_{\boldsymbol{y}}(X)} \boldsymbol{y}$ *is an optimal estimator of the ground-truth in the sense that it minimizes* $\epsilon'(X,h)$.

We omit the proof as it is straightforward (The mean minimizes mean squared error.). Theorem 1 makes sense intuitively: since $X$ is the only information we have, $\boldsymbol{y}$ can be any element in $\mathcal{U}_{\boldsymbol{y}}(X)$ and there is no information to support preferences of some elements over other elements in $\mathcal{U}_{\boldsymbol{y}}(X)$, so the best prediction one can make is the average of all elements in $\mathcal{U}_{\boldsymbol{y}}(X)$.

## 5 LEARNING THE HYPER LABEL MODEL

Although we have the analytical form of the optimal estimator $h^*$, computing it is of exponential complexity as $\mathcal{U}_{\boldsymbol{y}}(X)$ is exponentially large for any $X$. Therefore, we propose to train a neural network model $h$ to approximate the optimal estimator $h^*$. The trained model is a *hyper label model* that can infer the labels for a new dataset in a single forward pass. To materialize this idea, we need to answer the following questions: (1) What training data to use? (2) What model architecture to use? We discuss all these in the following sections.

### 5.1 TRAINING DATA GENERATION

Given a training set $\mathcal{D} = \{(X_1,\boldsymbol{y}_1),\dots\}$ and cross-entropy loss $\ell_{\text{CE}}(\cdot,\cdot)$, our learning objective is:

$$\arg\min_h \mathcal{L}(h,\mathcal{D}) = \arg\min_h \sum_{i=1}^{|\mathcal{D}|} \sum_{j=1}^{n} \ell_{\text{CE}}(h(X_i)[j],\boldsymbol{y}_i[j]), \tag{5}$$

where we use notation $[j]$ to index the $j$th item of the preceding vector. The key challenge is how to obtain the training dataset $\mathcal{D}$. Naively, we could generate a random $X$ and then use the analytical

method to find $\boldsymbol{y}^*$, which is however computationally intractable. Therefore, we design an efficient data generation method that ensures the model trained on our generated data still approximates $h^*$. By the following theorem, we show that given a $X$, uniformly sampling a valid $\boldsymbol{y}$, *i.e.*, $\boldsymbol{y} \in \mathcal{U}_{\boldsymbol{y}}(X)$, to compose the training dataset $\mathcal{D}$ ensures the learned hyper label model is asymptotically close to the analytical solution.

**Theorem 2.** $\forall X \in \mathcal{D}$, *if the corresponding* $\boldsymbol{y}$ *is uniformly sampled and valid, when* $|\mathcal{D}| \rightarrow +\infty$, *then* $\arg\min_h \mathcal{L}(h, \mathcal{D}) \rightarrow h^*(X) = \frac{1}{|\mathcal{U}_{\boldsymbol{y}}(X)|} \sum_{\boldsymbol{y} \in \mathcal{U}_{\boldsymbol{y}}(X)} \boldsymbol{y}$.

See proof in Appendix B. Based on the theorem, we derive the following training data generation method such that for every $X$, the corresponding $\boldsymbol{y}$ is uniformly sampled and valid. **Step 1:** We first randomly generate the shape of $X$, by randomly draw $m$ (and $n$) from a uniform distribution $[L_m, H_m]$ (and $[L_n, H_n]$). We provide details of how to choose $L_m, H_m, L_n$ and $H_n$ in Appendix E and empirically show the trained model generalizes very well outside of the given regions of shape (In fact, 13 datasets out from the 14 datasets we evaluate on are outside of the given regions of shape). **Step 2:** Given the shape of $X$, we then generate $X$ and the corresponding $\boldsymbol{y}$ with the values being sampled uniformly at random. **Step 3:** If $\sigma(X, \boldsymbol{y}) = 1$, we keep it as a training data point. This process (Step 1, 2, and 3) is repeated untill we obtain enough training data. Apparently, since $\boldsymbol{y}$ is generated uniformly, for any two different and valid vectors $\boldsymbol{y}_1$ and $\boldsymbol{y}_2$ with $\sigma(X, \boldsymbol{y}_1) = \sigma(X, \boldsymbol{y}_2) = 1$, the probability of generating $\boldsymbol{y}_1$ equals to the probability of generating $\boldsymbol{y}_2$, *i.e.* $p(\boldsymbol{y}|X)$ is uniform. The probability of generating a valid pair in one trial is about $0.2$ (see Appendix C).

## 5.2 MODEL ARCHITECTURE DESIGN

Notably, the input of the model $h$ is a matrix $X$ of size $n \times m$ and the output is a vector $\hat{\boldsymbol{y}}$ of size $n$. Thus, a reasonable instantiation of $h$ should satisfy the following three properties: **(1) Ability to accept arbitrary input size:** The number of data points $n$ and LFs $m$ can vary for different datasets. The model $h$ should be able to accept an input matrix $X$ of arbitrary size. **(2) Invariance to permutation of LFs:** Intuitively, randomly shuffling the LFs should not change the prediction of any data point. Formally, let $P_m$ denote one arbitrary permutation of the $m$ integers in $[0, m-1]$. Invariance to permutation of LFs means that $h(X[:, P_m]) = h(X)$, $\forall P_m$. **(3) Equivariance to permutation of data points:** Smilarily, randomly shuffling the data points should not change the prediction of each data point. Formally, equivariance to permutation of data points means that $h(X[P_n, :]) = h(X)[P_n]$, $\forall P_n$ where $P_n$ is defined similarly as $P_m$.

We argue that a graph neural network (GNN) is a good fit here since it can accept input graph of arbitrary size and is permutation equivariant to the nodes (Sanchez-Lengeling et al., 2021). Therefore, we attempt to represent the input matrix $X$ as a graph and then use a GNN for $h$ in order to satisfy the aforementioned properties. Specifically, the left-most matrix and graph in Figure 1 illustrate how we represent an input matrix of size $3 \times 2$ as a graph. Entry $X[i, j]$, the weak label of the $i^{th}$ data point provided by the $j^{th}$ LF, is represented as a node $V_{i,j}$ with value $X[i, j]$. There are two types of edges: solid yellow edge and dashed blue edge. Nodes from the same LF (*i.e.* same column in matrix $X$) are connected with solid yellow edges and nodes from the same data point (*i.e.* same row in matrix $X$) are connected with dashed blue edges. The graph representation $G$ loses no information as one can recover $X$ (or its permutation $X[P_n, P_m]$) from $G$.

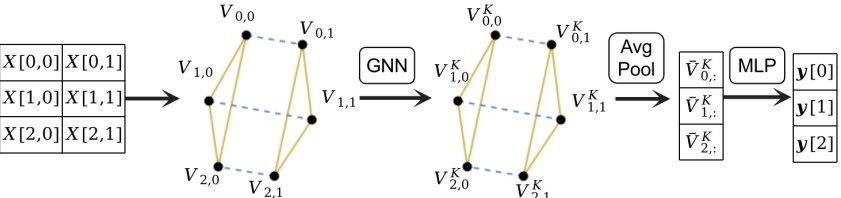

Figure 1: Overall network architecture.

In graph $G$, if we only look at dashed blue edges, there would be $n$ strongly connected components and each corresponds to one data point. Specifically, the strongly connected component $\text{SCC}_i = \{V_{i,0}, V_{i,1}, \dots\}$ corresponds to the $i^{th}$ data point. The overall model architecture is shown in Figure 1: first we encode the input graph with a GNN of $K$ layers where each node $V_{i,j}$ is encoded

with embedding $V_{i,j}^k$ at the $k^{th}$ layer; then after the final layer, we obtain an embedding for each SCC$_i$ (*i.e.* each data point) by pooling all of its nodes $\bar{V}_{i,:}^K = \frac{1}{m} \sum_j V_{i,j}^K$; The embedding of each SCC$_i$ is passed to a Multilayer perceptron (MLP) to obtain the final prediction. This architecture satisfies all three mentioned properties (see Appendix D.1).

We adopt the standard design of GNN. Since we have two types of edges, we perform message passing for neighboring nodes connected with each type of edges separately. Specifically, at the $k^{th}$ layer in the GNN, the embedding $V_{i,j}^k$ for the node $V_{i,j}$ is obtained as:

$$V_{i,j}^k = f_k(A^k(W_1^k \frac{1}{n} \sum_q V_{q,j}^{k-1}, W_2^k \frac{1}{m} \sum_q V_{i,q}^{k-1}, W_3^k \frac{1}{nm} \sum_{q,l} V_{q,l}^{k-1}, W_4^k V_{i,j}^{k-1})) \qquad (6)$$

where $W_1^k$, ... ,$W_4^k$ are weight matrices; $\frac{1}{n} \sum_q V_{q,j}^{k-1}$ denotes average pooling over neighboring nodes of $V_{i,j}$ connected with solid yellow edges and $\frac{1}{m} \sum_q V_{i,q}^{k-1}$ denotes average pooling over neighboring nodes of $V_{i,j}$ connected with dashed blue edges; Note we use average pooling because the graph can be of variable size as recommended by Sanchez-Lengeling et al. (2021) and we also include the node's previous embedding $V_{i,j}^{k-1}$ in the average in case the node has no neighbors (this is equivalent to adding a self-edge to each node.). We also add the global context of the graph $\frac{1}{nm} \sum_{j,j} V_{i,j}^{k-1}$ to enable message passing beyond neighboring nodes, following the standard practice (Gilmer et al., 2017; Battaglia et al., 2018); $A^k(\cdot, \cdot, \cdot, \cdot)$ denotes an aggregation operation and we use simple concatenation; $f_k$ denotes a linear layer with Relu activation.

**Handling Abstention.** Handling abstention is straightforward in our approach. We can simply remove the corresponding nodes in our graph. For example, when the $j^{th}$ LF abstains on the $i^{th}$ data point, we simply remove the node $V_{i,j}$ from the graph.

### 5.3 MODEL INFERENCE ON UNSEEN DATASET

The trained hyper label model can be applied to any new dataset with the inference being simply a single forward pass. During the forward pass, different data points (rows in matrix $X$) and different LFs (columns in $X$) exchange information through message passing in GNN. This information exchange step can be regarded as the counterpart of the dataset-specific training step of other methods.

**Inference Complexity.** The complexity of a forward pass is dominated by the GNN. Although there are $O(mn^2)$ edges in the graph, there is no need to actually materialize the $O(mn^2)$ edges and the complexity of each GNN layer is only $O(nm)$. In each GNN layer, for the three averaged pooling operations in Equation 6, the first one with complexity $O(n)$ needs to be computed once for each LF totaling $m$ times so the complexity is $O(nm)$; Similarly, the second one and the third one also have a complexity of $O(mn)$. Therefore, the time complexity for each GNN layer is $O(mn)$.

### 5.4 LEVERAGING GROUND TRUTH LABEL IF GIVEN

We pretrain a model $h_0$ on our sythetically generated data. When a small set of ground-truth labels is provided, our method can incorporate the labels by fine-tuning the model $h_0$ on the provided labels. Let $I$ denote the set of indices of the elements in $\boldsymbol{y}$ that are provided. For example, when $I = [2, 3]$, it means $\boldsymbol{y}[2]$ and $\boldsymbol{y}[3]$ are provided. Fine tuning is done by minimizing the cross-entropy loss $\sum_{i \in I} \ell_{CE}(h(X)[i], \boldsymbol{y}[i])$ and $h$ is initialized as the pretrained model $h_0$. After fine-tuning we obtain a model $h'$, and then all labels are obtained by $h'(X)$. We note the fine tuning process is dataset-specific, *i.e.* finetuning is done independently and specifically for each dataset $X$ using the ground-truth labels of that dataset.

### 5.5 SUPPORTING MULTI-CLASS CLASSIFICATION TASK

We have only considered the binary labels and it turns out that our trained model for binary labels can be easily used to support multi-class classification datasets by decomposing a multi-class task with $C$ classes to be $C$ one-vs-rest binary classification tasks. For multi-class tasks, we have $X[i, j] \in \{0, 1, 2, \dots C\}$ where 0 still denotes abstention and other numbers denote all the classes. We construct the label matrix for the $c^{th}$ class as $X_c[i, j] = 1$ if $X[i, j] = c$, $X_c[i, j] = 0$ if $X[i, j] = 0$, and otherwise $X_c[i, j] = -1$. In this way, we obtain $C$ label matrices $\{X_1, \dots X_c\}$.

We apply our pre-trained model $h_0$ on each label matrix of each class and obtain $C$ predicted probability vectors $(\boldsymbol{p_1}, \ldots, \boldsymbol{p_c})$. Then, for the $i^{th}$ data point, its soft label over the $C$ classes is $(\frac{\boldsymbol{p_1}[i]}{\sum_c \boldsymbol{p_c}[i]}, \ldots, \frac{\boldsymbol{p_c}[i])}{\sum_c \boldsymbol{p_c}[i]})$. We show in experiments this simple method works well on multi-class datasets (4 datasets out of the 14 datasets we use are multi-class datasets).

## 6 EXPERIMENTS

We evaluate the performance of all label models under both unsupervised and semi-supervised settings. We provide additional experimental results on the performance of end models trained on the generated labels by different label models in Appendix F.2. The code and instructions to reproduce the experiments are in supplementary materials.

**Datasets.** We use all 14 classification datasets in a recent weak supervision benchmark (Zhang et al., 2021) that are from diverse domains (e.g. income/sentiment/spam/relation/question/topic classification tasks). We highlight these datasets are only used for evaluation after our model is trained on synthetically generated data, and we never used these datasets during training. Table 1 shows the statistics of all datasets. We also use the metrics provided by the benchmark (Zhang et al., 2021) for each dataset (as different datasets need different metrics depending on their application background). All LFs are from the original authors of each dataset and are hosted in the benchmark project (Zhang, 2022a).

Table 1: 14 classification datasets from the weak supervision benchmark (Zhang et al., 2021)

| Dataset | Census | IMDB | Yelp | Youtube | SMS | Spouse | CDR | Commercial | Tennis | Basketball | AGNews | TREC | SemEval | ChemProt |
|---|---|---|---|---|---|---|---|---|---|---|---|---|---|---|
| #class | 2 | 2 | 2 | 2 | 2 | 2 | 2 | 2 | 2 | 2 | 4 | 6 | 9 | 10 |
| metric | F1 | acc | acc | acc | F1 | F1 | F1 | F1 | F1 | F1 | acc | acc | acc | acc |
| #LF | 83 | 5 | 8 | 10 | 73 | 9 | 33 | 4 | 6 | 4 | 9 | 68 | 164 | 26 |
| #Data | 31925 | 25000 | 38000 | 1956 | 5571 | 27766 | 14023 | 81105 | 8803 | 20256 | 120000 | 5965 | 2641 | 16075 |

We consider baselines for both unsupervised and semi-supervised label aggregation.

**Unsupervised Baselines:** *(1) Majority Vote (MV)*. The predicted label of each data point is the most common label given by LFs. *(2) Data Programming (DP)* (Ratner et al., 2016). DP uses a probabilistic graph model (PGM) where each LF is a node and the hidden ground truth is a latent variable. *(3) Flyingsquid (FS)* (Fu et al., 2020). FS also uses a PGM but gives a closed-form solution with some assumptions. *(4) MeTaL* (Ratner et al., 2019). MeTaL infers the ground truth using a matrix completion model. The latest version of the popular Snorkel system (snorkel team, 2022b) adopts MeTaL as its default label model. *(5) NPLM* (Yu et al., 2022). This method is also based on a PGM and assumes LFs are conditionally independent. It supports partial LFs that predict a subset of class labels and is designed to be very efficient. *(6) Dawid and Skene's method (DS)* (Dawid & Skene, 1979). DS models the confusion matrix of each LF with respect to the ground truth labels. The confusion matrix is learned by an Expectation-Maximization algorithm. *(7) Enhanced Bayesian Classifier Combination (EBCC)* (Li et al., 2019). This method models the joint distribution of LFs as a mixture of multiple low dimensional tensors. *(8) Constrained Label Learning (CLL)* (Arachie & Huang, 2021a). This method also minimizes the average prediction error where the error is defined using the unknown expected errors of each LFs. *(9) HLM*. This is our learned Hyper Label Model .
**Semi-supervised Baselines:** *(1) Semi-supervised DS* (Dawid & Skene, 1979). This is the semi-supervised extension of the Dawid and Skene's method. *(2) AMCL-CC (Mazzetto et al., 2021)*. This method uses labeled data to construct feasibility constraints and provides performance guarantees. *(3) Random Forest*. This method trains a random forest classifier with $X$ as the features using the provided labels. *(4) Semi-supervised HLM*. The semi-supervised version of our method HLM obtained by finetuning on the provided labels.

Note the baseline methods require a dataset-specific learning step, we use the transductive setting (data points used in unsupervised learning is also used to evaluate the learned model) following prior work (Mazzetto et al., 2021; Zhang, 2022b).

**Implementation.** We provide the implementation details of our method (e.g. setups and all parameters in data generation/model architecture/model training/validation) in Appendix E and implementation details of the experiments (e.g. hardware/datasets/baselines/setups) in Appendix G. Since

model training is important, here we provide a brief overview on training HLM. We generate each batch of data on-the-fly with a batch size of 50, *i.e.* each batch consists of 50 pairs of generated $(X, \boldsymbol{y})$. We train our model until training loss converges (loss doesn't decrease in $10^4$ iterations), which takes about $5 \times 10^4$ iterations. We noticed that in different runs, the performance of the trained model can vary, so we use a synthetic validation set $\mathcal{D}'$ to select the best run out of ten runs. The validation set is generated with a different method from a prior work (Zhang et al., 2021).

## 6.1 Unsupervised Label Aggregation

The performance of all methods on all 14 datasets averaged over five runs are shown in Table 2. To maintain the table to be readable, we only show the error bars for the averaged scores. Again, for our method HLM, we note only synthetically generated data is used for training and the 14 datasets are only used to evaluate the trained model. We note MV and our method HLM are deterministic while the other methods can give different results with different seeds. For HLM, the error bar is obtained by repeating the training process multiple times and then performing inference with different trained models.

**Main results.** First, our results align with the benchmark (Zhang et al., 2021) where MeTaL is slightly better than MV. The difference in numbers from the benchmark (Zhang et al., 2021) is due to that we use a transducive setting following (Mazzetto et al., 2021; Zhang, 2022b). Second, HLM outperforms the best baseline CLL by 1.4 points on average. Third, HLM is the best on 8 out of 14 datasets; On the remaining 6 datasets, HLM is the second best or is close to the second best method.

Table 2: Performance (F1 or acc score depending on the dataset) on all datasets

| Dataset | Census | IMDB | Yelp | Youtube | SMS | Spouse | CDR | Commercial | Tennis | Basketball | AGNews | TREC | SemEval | ChemProt | **AVG.** |
|---|---|---|---|---|---|---|---|---|---|---|---|---|---|---|---|
| MV | 22.2 | **75.0** | **74.4** | 80.3 | 84.0 | **51.6** | 63.3 | **85.9** | 85.0 | 18.9 | 81.4 | 49.9 | **84.2** | 53.7 | 65.0±0.0 |
| DP | 11.1 | 74.4 | 71.9 | 84.5 | 83.8 | 50.3 | 33.9 | 77.5 | **85.1** | 17.1 | 81.7 | 47.2 | 73.5 | **56.2** | 60.6±0.1 |
| FS | 17.1 | 74.5 | 74.0 | 83.7 | 74.4 | 49.9 | 69.6 | 82.5 | 84.0 | 17.1 | 81.3 | 50.1 | 23.8 | 52.4 | 59.6±0.0 |
| MeTaL | 51.1 | **75.0** | **74.4** | 86.0 | 57.7 | 49.9 | 67.9 | 83.7 | 80.9 | **19.0** | **82.2** | 52.1 | **84.2** | 52.9 | 65.5±0.2 |
| NPLM | 0.0 | 55.2 | 68.3 | 45.2 | 0.0 | 34.3 | 0.0 | 76.5 | 85.0 | 0.0 | 81.3 | 36.5 | 30.2 | 48.4 | 40.1±0.0 |
| DS | 0.0 | 74.4 | 68.3 | 45.2 | 65.0 | 34.3 | 0.1 | 77.8 | 85.0 | 17.1 | 26.6 | 20.9 | 73.5 | 35.1 | 44.5±0.0 |
| EBCC | 0.0 | 74.4 | 69.6 | 45.2 | 0.0 | 34.3 | 8.7 | 77.5 | 85.0 | 17.1 | 27.8 | 20.8 | 30.2 | 35.0 | 37.6±0.1 |
| CLL | 53.6 | 72.7 | 72.0 | 86.1 | **84.2** | 50.0 | 64.9 | 84.8 | 83.5 | 17.5 | 80.7 | 59.0 | **84.2** | 53.1 | 67.6±0.0 |
| HLM | **56.1** | **75.0** | **74.4** | **91.4** | 84.1 | **51.6** | **71.0** | 83.6 | 84.3 | 17.1 | 81.4 | **59.8** | **84.2** | 52.3 | **69.0**±0.2 |

**Efficiency.** We report the running time in Table 3. When measuring the running time, we use GPU for methods that support GPU (MeTaL, NPLM, and HLM). CPU-only running times are in Appendix F.1. HLM requires less than 1 seconds on every dataset. HLM is on average 6 times (and can be up to 18 times) faster than the fastest baseline (except Majority Vote) and is on average 50 times faster than the baseline (CLL) with the best accuracy. This is because all prior methods (except Majority Vote) require an unsupervised learning process while HLM performs prediction in a single forward pass just like Majority Vote. We note that these 14 benchmark datasets are relatively small (as creating a large benchmark dataset with ground-truth labels is expensive). In industry scenarios, LFs can be applied on millions of data points to create labels (Bach et al., 2019). The runtime gain of HLM will be more significant and HLM will enable the LF development process to be more interactive.

## 6.2 Semi-supervised Label Aggregation

For each dataset, we randomly sample $N_{\text{gt}}$ data points as the data points with known ground-truth labels and we evaluate on the remaining data points. When $N_{\text{gt}} > 0.7n$, we only select $0.7n$ data points to keep 30% of the data for evaluation in order to have a reliable evaluation score. We vary $N_{\text{gt}}$ from 10 to 10000. When finetuning HLM (with the method in Section 5.4), we use a smaller learning rate $lr = 0.0001$ to prevent overfitting (originally $lr = 0.001$). Intuitively, when $N_{\text{gt}}$ is small, we trust the pre-trained HLM more than the provided labels; when $N_{\text{gt}}$ is large, we trust the provided labels more than the pre-trained HLM. Therefore, we relate the number of finetuning epochs to $N_{\text{gt}}$ by setting the number of epochs as $\sqrt{N_{\text{gt}}}$.

Table 3: Running time (seconds) of label aggregation on all datasets

| Dataset | Census | IMDB | Yelp | Youtube | SMS | Spouse | CDR | Commercial | Tennis | Basketball | AGNews | TREC | SemEval | ChemProt | **AVG.** |
|---|---|---|---|---|---|---|---|---|---|---|---|---|---|---|---|
| MV | <0.1 | <0.1 | <0.1 | <0.1 | <0.1 | <0.1 | <0.1 | <0.1 | <0.1 | <0.1 | <0.1 | <0.1 | <0.1 | <0.1 | <0.1 |
| DP | 147.8 | 18.8 | 40.5 | 2.5 | 14.4 | 8.4 | 29.5 | 8.5 | 10.0 | 14.9 | 225.0 | 100.8 | 190.2 | 213.0 | 73.2 |
| FS | 21.1 | 1.7 | 3.7 | 0.2 | 3.2 | 0.8 | 3.7 | 0.6 | 0.6 | 14.9 | 22.1 | 16.3 | 69.0 | 26.4 | 12.2 |
| MeTaL | 0.5 | 0.3 | 0.4 | 0.4 | 0.4 | 0.3 | 0.4 | 0.4 | 0.4 | 0.4 | 0.5 | 3.6 | 4.6 | 3.6 | 1.2 |
| NPLM | 15.7 | 4.0 | 5.7 | 0.4 | 2.2 | 1.8 | 6.3 | 11.2 | 1.5 | 3.4 | 27.9 | 5.4 | 3.4 | 12.1 | 7.2 |
| DS | 2.4 | 79.8 | 116.1 | 0.2 | 3.6 | 0.9 | 29.7 | 267.7 | 4.6 | 2.1 | 16.3 | 78.3 | 36.6 | 255.9 | 63.9 |
| EBCC | 3.9 | 5.1 | 52.5 | 2.2 | 2.8 | 2.3 | 5.8 | 3.0 | 2.5 | 6.0 | 18.0 | 9.0 | 9.8 | 84.8 | 14.8 |
| CLL | 33.7 | 2.9 | 6.6 | 0.5 | 3.8 | 1.4 | 6.0 | 7.4 | 1.1 | 2.0 | 28.5 | 12.4 | 20.5 | 21.3 | 10.6 |
| HLM | 0.1 | 0.1 | 0.1 | 0.1 | 0.1 | 0.2 | 0.2 | 0.3 | 0.2 | 0.3 | 0.4 | 0.2 | 0.3 | 0.2 | 0.2 |

The results are shown in Figure 2. When $N_{\text{gt}} > 40$, semi-supervised HLM outperforms unsupervised HLM. Semi-supervised HLM outperforms the other three baselines when $N_{\text{gt}} < 1000$ and ties with AMCL-CC and Random Forest when $N_{\text{gt}} > 1000$. We highlight semi-supervised HLM is also the most efficient, e.g. when $N_{\text{gt}} = 10000$, the running time averaged over all datasets is 3.1 seconds for semi-supervised HLM, 61.3 seconds for Semi-supervised DS, 261.8 seconds for AMCL-CC, and 4.8 seconds for random forest.

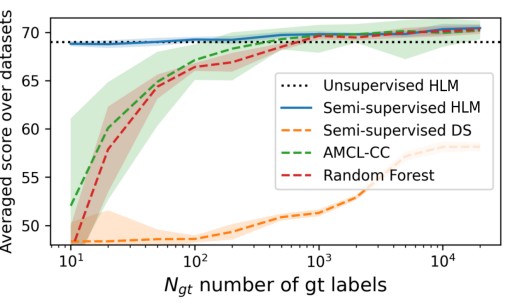

Figure 2: Semi-supervised performance.

Table 4: Ablation study. "*" denotes replacing the component it precedes with a naive one.

|  | Avg score |
|---|---|
| HLM | 69.0 |
| *data generation | 64.2 |
| *model architecture | 61.8 |
| *better-than-random | 67.6 |

## 6.3 ABLATION STUDY

We perform ablation study (under unsupervised label aggregation setting) in three aspects: (1) We replace our data generation method with the one proposed in (Zhang et al., 2021) that was originally used to generate LFs to evaluate label models. (2) We replace our model architecture with a naive architecture based on MLP and another architecture based on DeepSet (Zaheer et al., 2017) (see details in Appendix G). We report the best result of the two architectures. (3) We replace our better-than-random assumption in Equation 1 with a straightforward assumption that each LF is better than random in each class. The results are shown in Table 4. Replacing each component reduces performance. In particular, replacing our assumption with the straightforward assumption decreases performance because the assumption that each LF is better-than-random on each class is not satisfied in the real-world datasets.

## 7 CONCLUSION

We present a hyper label model for programmatic weak supervision, which infers the ground-truth labels for each dataset in a single forward pass and does not require any ad-hoc dataset-specific parameter learning step. The hyper label model approximates an analytical optimal method (which is computationally intractable due to its exponential complexity). We generate synthetic training data that ensures the trained hyper label model to approximate the analytical solution and design a model architecture based on GNN to ensure the model to be invariant to the permutation of LFs and equivariant to the permutation of data points. We experimentally verify the superiority of the hyper label model in both accuracy and efficiency with both unsupervised and semi-supervised label aggregation settings over 14 benchmark datasets.

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

## A  OPTIMALITIES OF THE UNIFORM DISTRIBUTION

We aim to approximate an unknown distribution $p(\boldsymbol{y}|X)$ (which can be different in different applications) with an fixed distribution $q(\boldsymbol{y}|X)$. Since both distributions are defined on a finite set $\mathcal{U}_{\boldsymbol{y}}(X)$, we can use the probabilities of the elements in $\mathcal{U}_{\boldsymbol{y}}(X)$ to represent each of the two distributions. Specifically, we represent $p(\boldsymbol{y}|X)$ as $\boldsymbol{p} = \{p_1, \ldots, p_{|\mathcal{U}_{\boldsymbol{y}}(X)|}\}$ and $q(\boldsymbol{y}|X)$ as $\boldsymbol{q} = \{q_1, \ldots, q_{|\mathcal{U}_{\boldsymbol{y}}(X)|}\}$. Similarly, we denote the uniform distribution (*i.e.* $p'(\boldsymbol{y}|X) = \frac{1}{|\mathcal{U}_{\boldsymbol{y}}(X)|}$) as $\boldsymbol{u} = \{u_1, \ldots u_{|\mathcal{U}_{\boldsymbol{y}}(X)|}\}$. Apparently, $\forall i$, we have $0 \le p_i \le 1$, $0 \le q_i \le 1$, $u_i = \frac{1}{|\mathcal{U}_{\boldsymbol{y}}(X)|}$, $\sum_i p_i = 1$, $\sum_i q_i = 1$, and $\sum_i u_i = 1$. Using the uniform distribution $\boldsymbol{u}$ to approximate $\boldsymbol{p}$ is the optimal in both the worst case and the average case. The two optimalities are formally defined as the following:

**(1) Worst-case Optimal:** The uniform distribution $\boldsymbol{u}$ has the minimum maximum distance to the unknown distribution $\boldsymbol{p}$:

$$\boldsymbol{u} = \underset{\boldsymbol{q}}{\arg\min} \max_{\boldsymbol{p}} \operatorname{dist}(\boldsymbol{p}, \boldsymbol{q}) \tag{7}$$

where "dist" can be the KL divergence or $L_\alpha$ distance $\forall \alpha > 1$. Note the conventional name is "$L_p$" distance, but to avoid reusing the same notation $p$ for different meanings, we use the name "$L_\alpha$" distance instead.

**(2) Average-case Optimal:** The uniform distribution $\boldsymbol{u}$ has the minimum expected KL divergence to the unknown distribution $\boldsymbol{p}$ under a mild assumption. Let $\mathcal{P}(\boldsymbol{p})$ denote the probability of the unknown distribution being a specific distribution $\boldsymbol{p}$ (e.g. $\mathcal{P}(\boldsymbol{u})$ would be denoting the probability of the unknown distribution being the uniform distribution *i.e.* $p(\boldsymbol{p} = \boldsymbol{u})$). Formally:

$$\boldsymbol{u} = \underset{\boldsymbol{q}}{\arg\min} E_{\boldsymbol{p}}[\operatorname{KL}(\boldsymbol{p}, \boldsymbol{q})] = \underset{\boldsymbol{q}}{\arg\min} \int_{\boldsymbol{p}} \operatorname{KL}(\boldsymbol{p}, \boldsymbol{q}) \mathcal{P}(\boldsymbol{p}) d\boldsymbol{p} \tag{8}$$

under the assumption that $\mathcal{P}(\boldsymbol{p})$ is centrally symmetric, formally:

$$\int_{\boldsymbol{p}} \mathcal{P}(\boldsymbol{p}) \boldsymbol{p} d\boldsymbol{p} = \boldsymbol{u} \tag{9}$$

We provide a formal proof for the two optimalities in the following:

**Proof for Worst-case Optimal.**

*Proof.* We first prove Equation 7 for KL divergence.

$$\max_{\boldsymbol{p}} \operatorname{KL}(\boldsymbol{p}, \boldsymbol{q})$$
$$= \max_{\boldsymbol{p}} \sum_i p_i \log \frac{p_i}{q_i} \tag{10}$$
$$= \max_{\boldsymbol{p}} \sum_i p_i \log p_i + p_i \log \frac{1}{q_i}$$

The maximum of the first term is zero, as $p_i \log p_i \le 0$ due to $p_i \ge 0$ and $\log p_i \le 0$. The maximum is obtained when there is a $j$ such that $p_j = 1$ and $p_i = 0$, $\forall i \ne j$;

$j$ also comes into play in the maximum of the second term. We have $\sum_i p_i \log \frac{1}{q_i} \le \sum_i p_i \max_k \log \frac{1}{q_k} = \max_k \log \frac{1}{q_k}$. Therefore, the maximum of of the second term is $\max_i \log \frac{1}{q_i}$ which is obtained when $j = \arg\max_i \log(\frac{1}{q_i})$, $p_j = 1$ and $p_i = 0$, $\forall i \ne j$.

We can see that the maximum of both terms is achieved at the same time with $j = \arg\max_i \log(\frac{1}{q_i})$, $p_j = 1$ and $p_i = 0$, $\forall i \ne j$. The maximum value is $\max_i \log \frac{1}{q_i}$.

Therefore,

$$\max_{\boldsymbol{p}} \operatorname{KL}(\boldsymbol{p}, \boldsymbol{q}) = \max_i \log \frac{1}{q_i}$$
$$= \log \frac{1}{\min_i q_i} \tag{11}$$
$$\ge \log |\mathcal{U}_{\boldsymbol{y}}(X)|$$

The inequality is because $\min_i q_i \leq \frac{1}{|\mathcal{U}_y(X)|}$ (otherwise, $\sum_i q_i > \sum_i \frac{1}{|\mathcal{U}_y(X)|} > 1$). The equality of the inequality is obtained when $\boldsymbol{q}$ is the uniform distribution, *i.e.* $\boldsymbol{q} = \boldsymbol{u}$. Therefore, $\boldsymbol{u} = \arg\min_{\boldsymbol{q}} \max_{\boldsymbol{p}} \mathrm{KL}(\boldsymbol{p}, \boldsymbol{q})$.

Next, we prove Equation 7 for $L_\alpha$ distance $\forall \alpha > 1$.

The $L_\alpha$ distance is defined as:

$$L_\alpha(\boldsymbol{p}, \boldsymbol{q}) = (\sum_i |p_i - q_i|^\alpha)^{1/\alpha} \tag{12}$$

Take the derivative of $L_\alpha(\boldsymbol{p}, \boldsymbol{q})$ with respect to a $p_i$:

$$\frac{\partial L_\alpha(\boldsymbol{p}, \boldsymbol{q})}{p_i} = \begin{cases} \frac{1}{\alpha}(\sum_i |p_i - q_i|^\alpha)^{1/\alpha - 1}\alpha(p_i - q_i)^{\alpha - 1} \text{ if } p_i \geq q_i \\ -\frac{1}{\alpha}(\sum_i |p_i - q_i|^\alpha)^{1/\alpha - 1}\alpha(q_i - p_i)^{\alpha - 1} \text{ otherwise} \end{cases} \tag{13}$$

This means if $p_i - q_i \geq p_j - q_j$, $\frac{\partial L_\alpha(\boldsymbol{p}, \boldsymbol{q})}{p_i} \geq \frac{\partial L_\alpha(\boldsymbol{p}, \boldsymbol{q})}{p_j}$. Therefore, replacing $p_i, p_j$ with $p_i + \delta$, $p_j - \delta$ where $\delta > 0$ increases $L_\alpha(\boldsymbol{p}, \boldsymbol{q})$ and eventually replacing $p_i, p_j$ with $p_i + p_j$, 0 increases $L_\alpha(\boldsymbol{p}, \boldsymbol{q})$. Let $k = \arg\max_i p_i - q_i$. For each pair $(p_k, p_i)i \neq k$, we replace $p_k$ to be $p_k + p_i$ and $p_i$ to be 0 and eventually we have $p_k = 1$ and $p_i = 0, i \neq k$:

$$(\sum_i |p_i - q_i|^\alpha)^{1/\alpha} \leq (\sum_{i \neq k} |q_i|^\alpha + |1 - q_k|^\alpha)^{1/\alpha} \tag{14}$$

Apparently, when $k = \arg\min_i q_k$, the right hand side is further maximized. Without loss of generality, we can assume $q_1 \leq q_2 \leq \cdots \leq q_{|\mathcal{U}_y(X)|}$. Therefore:

$$\max_{\boldsymbol{p}} L_\alpha(\boldsymbol{p}, \boldsymbol{q}) = ((1 - q_1)^\alpha + \sum_{i>1} q_i^\alpha)^{1/\alpha} \tag{15}$$

By the Hölder's inequality (Hardy et al., 1988):

$$\sum_{i>1} q_i^\alpha \geq (|\mathcal{U}_y(X)| - 1)^{1-\alpha}(\sum_{i>1} q_i)^\alpha = (|\mathcal{U}_y(X)| - 1)^{1-\alpha}(1 - q_1)^\alpha \tag{16}$$

where equality in the inequality is obtained when $q_2 = \cdots = q_{|\mathcal{U}_y(X)|}$. Therefore:

$$\begin{aligned} \max_{\boldsymbol{p}} L_\alpha(\boldsymbol{p}, \boldsymbol{q}) &\geq ((1 - q_1)^\alpha + (|\mathcal{U}_y(X)| - 1)^{1-\alpha}(1 - q_1)^\alpha)^{1/\alpha} \\ &\geq ((1 - \frac{1}{|\mathcal{U}_y(X)|})^\alpha + (|\mathcal{U}_y(X)| - 1)^{1-\alpha}(1 - \frac{1}{|\mathcal{U}_y(X)|})^\alpha)^{1/\alpha} \\ &= ((\frac{|\mathcal{U}_y(X)| - 1}{|\mathcal{U}_y(X)|})^\alpha + \frac{|\mathcal{U}_y(X)| - 1}{|\mathcal{U}_y(X)|^\alpha})^{1/\alpha} \end{aligned} \tag{17}$$

where the second inequality is because $((1-q_1)^\alpha + (|\mathcal{U}_y(X)| - 1)^{1-\alpha}(1-q_1)^\alpha)^{1/\alpha}$ monotonically decreases as $q_1$ increases and we have $q_1 \leq \frac{1}{|\mathcal{U}_y(X)|}$ because $q_1$ is the minimum in $\boldsymbol{q}$, *i.e.* $q_1 \leq q_2 \leq \cdots q_{|\mathcal{U}_y(X)|}$.

In summary, the minimum of $\max_{\boldsymbol{p}} L_\alpha(\boldsymbol{p}, \boldsymbol{q})$ is obtained when $q_2 = q_3 = \cdots = q_{|\mathcal{U}_y(X)|}$ and $q_1 = \frac{1}{|\mathcal{U}_y(X)|}$, which means $q_1 = q_2 = q_3 = \cdots = q_{|\mathcal{U}_y(X)|} = \frac{1}{|\mathcal{U}_y(X)|}$. In other words, $\boldsymbol{q} = \boldsymbol{u}$. Therefore, $\boldsymbol{u} = \arg\min_{\boldsymbol{q}} \max_{\boldsymbol{p}} L_\alpha(\boldsymbol{p}, \boldsymbol{q})$

$\square$

**Proof for Average-case Optimal.**

*Proof.*

$$\begin{aligned} E_{\boldsymbol{p}}[\mathrm{KL}(\boldsymbol{p}, \boldsymbol{q})] &= \int_{\boldsymbol{p}} \mathrm{KL}(\boldsymbol{p}, \boldsymbol{q})\mathcal{P}(\boldsymbol{p})d\boldsymbol{p} \\ &= \int_{\boldsymbol{p}} \mathcal{P}(\boldsymbol{p}) \sum_i p_i \log \frac{p_i}{q_i} d\boldsymbol{p} \\ &= \int_{\boldsymbol{p}} \mathcal{P}(\boldsymbol{p}) \sum_i p_i \log p_i d\boldsymbol{p} - \int_{\boldsymbol{p}} \mathcal{P}(\boldsymbol{p}) \sum_i p_i \log q_i d\boldsymbol{p} \end{aligned} \tag{18}$$

Since the first term is irrelevant to $\boldsymbol{q}$, we have:

$$
\begin{aligned}
E_{\boldsymbol{p}}[\mathrm{KL}(\boldsymbol{p}, \boldsymbol{q})] &= \text{constant} - \int_{\boldsymbol{p}} \mathcal{P}(\boldsymbol{p}) \sum_i p_i \log q_i d\boldsymbol{p} \\
&= \text{constant} - \sum_i \log q_i \int_{\boldsymbol{p}} \mathcal{P}(\boldsymbol{p}) p_i d\boldsymbol{p} \\
&= \text{constant} - \sum_i u_i \log q_i
\end{aligned}
\tag{19}
$$

where the last equation is by the assumption that $\mathcal{P}(\boldsymbol{p})$ is centrally symmetric, *i.e.* $\int_{\boldsymbol{p}} \mathcal{P}(\boldsymbol{p}) \boldsymbol{p} d\boldsymbol{p} = \boldsymbol{u}$. Therefore:

$$
\begin{aligned}
E_{\boldsymbol{p}}[\mathrm{KL}(\boldsymbol{p}, \boldsymbol{q})] &= \text{constant} - \sum_i u_i \log q_i \\
&= \text{constant} - \frac{1}{|\mathcal{U}_{\boldsymbol{y}}(X)|} \sum_i \log q_i \\
&= \text{constant} - \frac{1}{|\mathcal{U}_{\boldsymbol{y}}(X)|} \log \prod_i q_i \\
&\geq \text{constant} - \frac{1}{|\mathcal{U}_{\boldsymbol{y}}(X)|} \log\left(\left(\frac{\sum_i q_i}{|\mathcal{U}_{\boldsymbol{y}}(X)|}\right)^{|\mathcal{U}_{\boldsymbol{y}}(X)|}\right) \\
&= \text{constant} + \log(|\mathcal{U}_{\boldsymbol{y}}(X)|)
\end{aligned}
\tag{20}
$$

where the inequality is the inequality of arithmetic and geometric means. The equality of the inequality is obtained when $q_1 = q_2 = \cdots = q_{|\mathcal{U}_{\boldsymbol{y}}(X)|} = \frac{1}{|\mathcal{U}_{\boldsymbol{y}}(X)|}$, *i.e.* $\boldsymbol{q} = \boldsymbol{u}$. Therefore, $\boldsymbol{u} = \arg\min_{\boldsymbol{q}} E_{\boldsymbol{p}}[\mathrm{KL}(\boldsymbol{p}, \boldsymbol{q})]$. $\square$

## B  PROOF FOR THEOREM 2

$\forall X \in \mathcal{D}$, if the corresponding $\boldsymbol{y}$ is uniformly sampled and valid, when $|\mathcal{D}| \to +\infty$, then $\arg\min_h \mathcal{L}(h, \mathcal{D}) \to h^*(X) = \frac{1}{|\mathcal{U}_{\boldsymbol{y}}(X)|} \sum_{\boldsymbol{y} \in \mathcal{U}_{\boldsymbol{y}}(X)} \boldsymbol{y}$.

*Proof.* For each $X$, let $\mathcal{D}(X)$ denote the subset $\{(X, \boldsymbol{y}'_1), (X, \boldsymbol{y}'_2), \dots\}$ of $\mathcal{D}$. The cross entropy loss on $\mathcal{D}(X)$ is:

$$
-\sum_{i=1}^{|\mathcal{D}(X)|} \sum_{j=1}^{n} \frac{1 + \boldsymbol{y}'_i[j]}{2} \log\left(\frac{1 + h(X)[j]}{2}\right) + \left(1 - \frac{1 + \boldsymbol{y}'_i[j]}{2}\right) \log\left(1 - \frac{1 + h(X)[j]}{2}\right)
\tag{21}
$$

where $n$ denotes the number of rows in $X$; We use $[j]$ to index the $j$th item of its preceding vector; We convert the region of $\boldsymbol{y}'_i[j]$ and $h(X)[j]$ from $[-1, 1]$ to $[0, 1]$ by adding 1 then dividing by 2. By taking derivative and setting it to zero, the above equation is minimized when:

$$
h(X)[j] = \frac{\sum_{i=1}^{|\mathcal{D}(X)|} \boldsymbol{y}'_i[j]}{|\mathcal{D}(X)|}, \ \forall j
\tag{22}
$$

When $|\mathcal{D}| \to +\infty$ (so that $|\mathcal{D}(X)| \to +\infty$), by the law of large numbers (Dekking et al., 2005), $h(X)[j] = \frac{\sum_{i=1}^{|\mathcal{D}(X)|} \boldsymbol{y}'_i[j]}{|\mathcal{D}(X)|} = E(\boldsymbol{y}[j]|X)$. Since $p(\boldsymbol{y}|X)$ is uniform, $E(\boldsymbol{y}[j]|X) = \sum_{\boldsymbol{y} \in \mathcal{U}_{\boldsymbol{y}}(X)} \frac{1}{|\mathcal{U}_{\boldsymbol{y}}(X)|} \boldsymbol{y}[j]$ for $\forall j$. This means $h(X) = \sum_{\boldsymbol{y} \in \mathcal{U}_{\boldsymbol{y}}(X)} \frac{1}{|\mathcal{U}_{\boldsymbol{y}}(X)|} \boldsymbol{y} = h^*(X)$. $\square$

## C  PROBABILITY OF GENERATING A VALID PAIR

To simplify our analysis, in the following, we only consider $\boldsymbol{y}$ that contains both $-1$ and $+1$, which has a probability $p_0 = 1 - \frac{2}{2^n}$. $p_0 \approx 1$ when $n \geq 100$ (When generating data, we sample $n$ from $[L_n, H_n] = [100, 2000]$ which we explain in Appendix E).

Let $S$ denote the set of all possible pairs of $(X, \boldsymbol{y})$ and let $\mathcal{U} = \{(X, \boldsymbol{y}) | \sigma(X, \boldsymbol{y}) = 1\}$ denote the set of all valid pairs. $S$ is made up by three subsets: $\mathcal{U} = \{(X, \boldsymbol{y}) | \sigma(X, \boldsymbol{y}) = 1\}$, $S_e = \{(X, \boldsymbol{y}) | \sum_{j=0}^{m-1} g(X, \boldsymbol{y}, j, -1) = \frac{m}{2}$ or $\sum_{j=0}^{m-1} g(X, \boldsymbol{y}, j, 1) = \frac{m}{2}\}$ and $S_c = S - \mathcal{U} - S_e$. Apparently $S_c$ is also made up by three subsets, *i.e.* $S_c = S_{c_1} \cup S_{c_2} \cup S_{c_3}$ where $S_{c_1} = \{(X, \boldsymbol{y}) | \sum_{j=0}^{m-1} g(X, \boldsymbol{y}, j, -1) < \frac{m}{2}$ and $\sum_{j=0}^{m-1} g(X, \boldsymbol{y}, j, 1) < \frac{m}{2}\}$, $S_{c_2} = \{(X, \boldsymbol{y}) | \sum_{j=0}^{m-1} g(X, \boldsymbol{y}, j, -1) < \frac{m}{2}$ and $\sum_{j=0}^{m-1} g(X, \boldsymbol{y}, j, 1) > \frac{m}{2}\}$ and $S_{c_3} = \{(X, \boldsymbol{y}) | \sum_{j=0}^{m-1} g(X, \boldsymbol{y}, j, -1) > \frac{m}{2}$ and $\sum_{j=0}^{m-1} g(X, \boldsymbol{y}, j, 1) < \frac{m}{2}\}$.

**Lemma 3.** $|S_{c_1}| = |S_{c_2}| = |S_{c_3}| = |\mathcal{U}|$.

*Proof.* For each element $(X, \boldsymbol{y})$ in $S_{c_1}$, $\sum_{j=0}^{m-1} g(X, \boldsymbol{y}, j, -1) < \frac{m}{2}$ and $\sum_{j=0}^{m-1} g(X, \boldsymbol{y}, j, 1) < \frac{m}{2}$. We can flip $X[\boldsymbol{y} = 1, :]$ to be $-X[\boldsymbol{y} = 1, :]$ and flip $X[\boldsymbol{y} = -1, :]$ to be $-X[\boldsymbol{y} = -1, :]$. After flipping, we obtain pair $(X', \boldsymbol{y})$, and apparently $(X', \boldsymbol{y}) \in \mathcal{U}$. This means for each element in $S_{c_1}$ there is a corresponding element in $\mathcal{U}$, so we have $|S_{c_1}| \leq |\mathcal{U}|$. Similarly, for each element in $\mathcal{U}$, we can do flipping to get an element in $S_{c_1}$, so we also have $|\mathcal{U}| \leq |S_{c_1}|$. Therefore, $|\mathcal{U}| = |S_{c_1}|$. Similarly, one can show that $|\mathcal{U}| = |S_{c_2}|$ and $|\mathcal{U}| = |S_{c_3}|$. □

By Lemma 3, $|S_c| = |S_{c_1}| + |S_{c_2}| + |S_{c_3}| = 3|\mathcal{U}|$. When $m$ is odd, apparently, $|S_e| = 0$. Therefore, the probability of a randomly generated pair being valid is:

$$p((X, \boldsymbol{y}) \in \mathcal{U}|m) = \frac{|\mathcal{U}|}{|S|} = \frac{|\mathcal{U}|}{|\mathcal{U}| + |S_e| + |S_c|} = \frac{1}{4} \tag{23}$$

Next, we consider when $m$ is even. To simplify our analysis, approximately, $p(g(X, \boldsymbol{y}, j, -1)) = \frac{1}{2}$ and $p(g(X, \boldsymbol{y}, j, 1)) = \frac{1}{2}$. This is because the probability that the number of correct elements exactly equal to the number of incorrect elements for each class is extremely small due to $n$ being relatively large. Therefore, we have:

$$p((X, \boldsymbol{y}) \in S_e|m) = \frac{\binom{m}{m/2}}{2^m} \tag{24}$$

Therefore:

$$p((X, \boldsymbol{y}) \in \mathcal{U}|m) = (1 - \frac{\binom{m}{m/2}}{2^m}) \frac{1}{4} \tag{25}$$

Since $m$ is uniformly sampled from $[L_m, H_m] = [2, 60]$ (which we explain in Appendix E), we have:

$$p((X, \boldsymbol{y}) \in \mathcal{U}) = \sum_{m, m\%2=1, L_m \leq m \leq H_m} \frac{1}{H_m - L_m} \frac{1}{4} + \sum_{m, m\%2=0, L_m \leq m \leq H_m} \frac{1}{H_m - L_m} (1 - \frac{\binom{m}{m/2}}{2^m}) \frac{1}{4}$$

$$= \frac{1}{2} \times \frac{1}{4} + \sum_{m, m\%2=0, L_m \leq m \leq H_m} \frac{1}{H_m - L_m} (1 - \frac{\binom{m}{m/2}}{2^m}) \frac{1}{4}$$

$$\approx 0.232$$

$$\tag{26}$$

This means the probability of generating a valid pair in one trial is about 0.232.

## D  DISCUSSIONS

### D.1  THE PROPOSED ARCHITECTURE SATISFIES THE THREE PROPERTIES

To see how the proposed architecture in Figure 1 satisfies the three properties mentioned in the begining of Section 5.2. First, GNN accepts arbitrary input size, so $X$ can be of any size; Second, GNN is permutation equivariant to the nodes, so the output embeddings of GNN are equivariant to the permutation of data points and LFs. After average pooling for each data point over all LFs (each SCC with dashed blue edges), the network is invariant to the permutation of LFs and is still equivariant to the permutation of data points.

### D.2 Crowdsourcing Methods for Weak Supervision

The two crowdsourcing methods have the worst performance in Table 2. The reason that crowdsourcing methods don't work well on weak supervision datasets has not been investigated or discussed in prior work, and we provide our conjecture. First, the label matrix in crowdsourcing tends to be extremely sparse as there can be many crowd workers while each crowd worker might annotate a few data points then quit (Zheng et al., 2017); In contrast, in weak supervision, each LF is applied to each data point. Second, since crowd workers are humans, the labels provided by the crowd workers tend to have higher accuracy; In contrast, a LF when applied on data unseen by the LF developer can predict very noisy labels. In other words, the existing crowdsourcing methods are designed to work in the sparse scenario with weak labels of higher accuracy, so that they don't work well in the weak supervision setting with a denser and noisier label matrix.

## E  Implementation Details of HLM

**Data Generation.** When generating each pair $(X, \boldsymbol{y})$, we first randomly generate $n$ and $m$, the number of rows/columns of matrix $X$. Note $n$ is the number of data points and $m$ is the number of LFs. As we mentioned, we first sample $n$ and $m$ uniformly from $[L_n, H_n]$ and $[L_m, H_m]$ respectively. We set $[L_n, H_n] = [100, 2000]$ where $L_n = 100$ is because typically there are at least hundreds of data points otherwise it is not necessary to write LFs as one can just manually label all data points and we set $H_n = 2000$ due to memory limit during model training. We set $[L_m, H_m] = [2, 60]$ where $L_m = 2$ is because when there is only one LF there is no need to aggregate and we set $H_m = 60$ due to memory limit during model training; We highlight our trained model generalizes well to number of LFs and number of data points (see Table 1) that are not in the region $[L_m, H_m]$ and $[L_n, H_n]$ as we have shown in experiments. Once we have $n$ and $m$, we invoke the method mentioned in Section 5.1 to generate $(X, \boldsymbol{y})$.

Since our data is synthetically generated, there is no need to generate a fixed training set. Our training data is generated on the fly, *i.e.* during training when the data loader fetches the next pair of $(X, \boldsymbol{y})$, a new pair is immediately generated and returned.

**Model Architecture.** We implement our model architecture in Pytorch (Paszke et al., 2019). We use $K = 4$ layers of GNN. The embedding dimension of GNN is 32, *i.e.* each node in the graph is encoded with a 32 dimensional embedding. The final MLP consists of three linear layers; the first two linear layers use Relu activation and the last linear layer uses Sigmoid activation.

**Model Training.** We use the Adam optimizer (Kingma & Ba, 2014). We set amsgrad to be true for better convergence (Reddi et al., 2019) and keep all other parameters as the default values provided by Pytorch (e.g. learning rate $lr = 0.001$). We use a batch size of 50, *i.e.* each batch consists of 50 pairs of generated $(X, \boldsymbol{y})$. We tested different batch sizes of 16 and 250 and observed no meaningful difference. We train our model until training loss converges (loss doesn't decrease in $10^4$ iterations), which takes about one day with $5 \times 10^4$ iterations on a K80 GPU. Note one iteration means one gradient update/one batch, and we don't have the notion of epoch as training data is generated on-the-fly for each batch.

**Validation.** We also need to prevent our model from overfitting the training set. We highlight that, different from typical ML settings where one gets access to a validation set that is similar to the test set, in our setting we have no validation set that is similar to the test set. Again, when training our model, the real test datasets are unseen and we only have access to synthetic data. Our intuition is that when the model overfits the sythetically generated training set $\mathcal{D}$, its performance will be poor on data that is different from the training set, for example, on another sythetic dataset $\mathcal{D}'$ that is generated in a different way. We synthetically generate the validation set $\mathcal{D}'$ with size $|\mathcal{D}'| = 100$ according to the generation method proposed in (Zhang et al., 2021); In this method, LFs are independent from each other conditioned on the ground-truth label.

We note that the way we use the validation set is also different from a typical setting. We train the model until training loss converges (this typically requires about $5 \times 10^4$ iterations), and repeat 10 runs (*i.e.* train our model 10 times from scratch). We then select the run with the highest averaged validation accuracy over all iterations (as validation accuracy might fluctuate over iterations); We use the learned model at the final iteration of the selected run in our experiments. We provide our

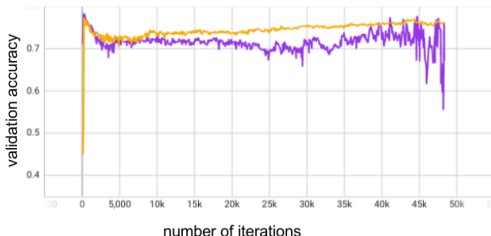

Figure 3: Accuracy on the synthetic validation set $\mathcal{D}'$ vs number of training iterations. The purple line and yellow line are two different runs. The yellow run is selected as it is more stable with a higher averaged validation accuracy.

reasoning of doing this: (1) We do not use the validation set to do early stopping (*i.e.* to select the best iteration in a run). In a typical ML setting, the validation set is used to select the best epoch/iteration. This is possible because in a typical ML setting the validation set is similar to the test set and the validation set provides very strong signal towards which iteration is a good iteration for the test set. In our case, the validation set $\mathcal{D}'$ can be very different from the test set, thus the selected iteration based on $\mathcal{D}'$ might not be a good iteration for the test set. (2) We use the validation set to select the best run. We observed that at different runs, the curve of validation accuracy vs number of iteration can be different (e.g. the two runs in Figure 3), so the test accuracy of the model in different runs can be different. We would like to select the best run using the validation set $\mathcal{D}'$. Intuitively, one run with better validation accuracy on average over all iterations is stably better (e.g. the yellow run in Figure 3), so we select the run with an best averaged validation accuracy over iterations. As an example, for the two runs in Figure 3, although the highest validation accuracy of purple run can be higher than that of the yellow run, the yellow run has a higher averaged validation accuracy over iterations and is much more stable, so we select the yellow run. We also observed this run to have a less degree of fluctuation in validation accuracy, as shown in Figure 3. This suggests the model converges at a flat minima, which is known to generalize better (Li et al., 2018; Keskar et al., 2016; Izmailov et al., 2018).

One natural question is that why it is possible to select the best run but it is not possible to select the best iteration. The reason is that selecting the best run out from 10 runs require much less information than selecting the best iteration out from $5 \times 10^4$ iterations. Since the validation set $\mathcal{D}'$ can be very different from the test set, the information provided by $\mathcal{D}'$ is very limited.

An interesting phenomenon in the validation accuracy curve in Figure 3 for the yellow run is that validation accuracy first increases then decreases and finally increases. A similar trend was observed in prior work (of a different task) that also trains a model on synthetic data and validate on a different data distribution (Wu et al., 2022c). We believe this is a double descent phenomenon (Nakkiran et al., 2021) induced by the distributional difference between the training and validation sets.

## F   ADDITIONAL EXPERIMENT RESULTS

### F.1   RUNNING TIME

We report the running time for all methods in Table 5. For MV, DP, FS, DS, EBCC, and CLL, the running is on CPU as these methods do not support GPU. For MeTaL, NPLM and HLM, we report the running time on CPU and GPU.

### F.2   END MODEL PERFORMANCE

We use the generated labels of each method to train an end model for each dataset. We consider the three best performing baselines MV, MeTaL and CLL . We use the test split provided by the benchmark (Zhang et al., 2021) for each dataset because some datasets only have ground-truth labels for data points in the provided test split. We then randomly split the remaining data points to be a training set and a validation set with a 3:1 ratio. The labels in the training set and validation set are generated labels by each label model, while the labels in the test set are ground-truth labels for evaluation. Following prior work (Ratner et al., 2016; Zhang et al., 2021), the probabilistic labels

Table 5: Running time (seconds) of label aggregation on all datasets with CPU and GPU.

| Dataset | Census | IMDB | Yelp | Youtube | SMS | Spouse | CDR | Commercial | Tennis | Basketball | AGNews | TREC | SemEval | ChemProt | AVG. |
|---|---|---|---|---|---|---|---|---|---|---|---|---|---|---|---|
| MV | <0.1 | <0.1 | <0.1 | <0.1 | <0.1 | <0.1 | <0.1 | <0.1 | <0.1 | <0.1 | <0.1 | <0.1 | <0.1 | <0.1 | <0.1 |
| DP | 147.8 | 18.8 | 40.5 | 2.5 | 14.4 | 8.4 | 29.5 | 8.5 | 10.0 | 14.9 | 225.0 | 100.8 | 190.2 | 213.0 | 73.2 |
| FS | 21.1 | 1.7 | 3.7 | 0.2 | 3.2 | 0.8 | 3.7 | 0.6 | 0.6 | 14.9 | 22.1 | 16.3 | 69.0 | 26.4 | 12.2 |
| MeTaL-GPU | 0.5 | 0.3 | 0.4 | 0.4 | 0.4 | 0.3 | 0.4 | 0.4 | 0.4 | 0.4 | 0.5 | 3.6 | 4.6 | 3.6 | 1.2 |
| MeTaL-CPU | 1.1 | 0.3 | 0.4 | 0.4 | 0.9 | 0.3 | 0.9 | 0.4 | 0.4 | 0.4 | 0.5 | 16.7 | 18.1 | 16.5 | 4.1 |
| NPLM-GPU | 15.7 | 4.0 | 5.7 | 0.4 | 2.2 | 1.8 | 6.3 | 11.2 | 1.5 | 3.4 | 27.9 | 5.4 | 3.4 | 12.1 | 7.2 |
| NPLM-CPU | 156.7 | 56.0 | 5.7 | 0.4 | 19.8 | 1.8 | 6.3 | 11.2 | 1.5 | 3.4 | 29.9 | 49.9 | 25.9 | 132.6 | 35.8 |
| DS | 2.4 | 79.8 | 116.1 | 0.2 | 3.6 | 0.9 | 29.7 | 267.7 | 4.6 | 2.1 | 16.3 | 78.3 | 36.6 | 255.9 | 63.9 |
| EBCC | 3.9 | 5.1 | 52.5 | 2.2 | 2.8 | 2.3 | 5.8 | 3.0 | 2.5 | 6.0 | 18.0 | 9.0 | 9.8 | 84.8 | 14.8 |
| CLL | 33.7 | 2.9 | 6.6 | 0.5 | 3.8 | 1.4 | 6.0 | 7.4 | 1.1 | 2.0 | 28.5 | 12.4 | 20.5 | 21.3 | 10.6 |
| HLM-GPU | 0.1 | 0.1 | 0.1 | 0.1 | 0.1 | 0.2 | 0.2 | 0.3 | 0.2 | 0.3 | 0.4 | 0.2 | 0.3 | 0.2 | 0.2 |
| HLM-CPU | 0.9 | 0.2 | 0.3 | 0.2 | 0.2 | 0.7 | 0.7 | 1.1 | 0.3 | 0.9 | 4.6 | 4.4 | 1.9 | 7.3 | 1.7 |

instead of the hard labels are used to train the end model when possible. We adopt the end models used in (and their implementations provided by) the benchmark (Zhang et al., 2021; Zhang, 2022a), *i.e.* a pretrained BERT model (Devlin et al., 2018) for textual datasets and a multi-layer perception (MLP) for datasets with numeric features. We report the results on test set in Table 6. Again, to maintain the table to be readable, we only show the error bars for the averaged scores.

Table 6: Performance of end model trained with labels generated by each method.

| Dataset | Census | IMDB | Yelp | Youtube | SMS | Spouse | CDR | Commercial | Tennis | Basketball | AGNews | TREC | SemEval | ChemProt | AVG. |
|---|---|---|---|---|---|---|---|---|---|---|---|---|---|---|---|
| End model | MLP | BERT | BERT | BERT | BERT | BERT | BERT | MLP | MLP | MLP | BERT | BERT | BERT | BERT | |
| MV | 31.7 | **74.7** | 74.2 | 90.9 | 83.5 | 51.6 | 62.9 | **90.1** | 83.5 | 13.4 | 81.9 | 63.9 | 76.8 | **54.6** | 66.7±0.8 |
| MeTaL | 11.6 | 74.4 | 70.5 | 87.1 | **87.3** | 51.0 | 63.5 | 88.2 | **83.5** | 14.5 | **82.0** | 63.7 | **83.1** | 52.5 | 65.2±0.2 |
| CLL | 51.5 | 73.6 | 72.8 | 81.8 | 83.6 | 52.0 | 59.8 | 89.6 | **83.5** | **19.3** | 81.7 | **68.4** | 82.8 | 53.0 | 68.1±0.3 |
| HLM | **56.3** | **74.7** | **75.7** | **93.0** | 82.4 | **52.3** | **64.1** | 87.1 | **83.5** | 17.0 | 80.8 | **68.4** | 82.8 | 53.1 | **69.4**±0.3 |

Our results align with those in the benchmark (Zhang et al., 2021) where the end model trained on labels generated by MeTaL is slightly worse than that by MV. Overall, HLM outperforms the other three methods. On Yelp, Spouse, and SemEval, HLM tied with MV in label quality (see Table 2) but has better end model performance as HLM's probabilistic labels can be more informative. Note the scores of the end model can be higher than that of the generated labels (as also observed in the benchmark (Zhang et al., 2021) and prior work (Ratner et al., 2017)) because the end model incorporates additional information from the raw data.

## G IMPLEMENTATION DETAILS OF EXPERIMENTS

**Hardware.** All of our experiments were performed on a machine with a 2.20GHz Intel Xeon(R) Gold 5120 CPU, a K80 GPU and with 96GB 2666MHz RAM.

**Datasets.** We use the datasets prepared by the wrench benchmark on Github (Zhang, 2022a; Zhang et al., 2021). All the datasets and LFs are publicly released by previous work (Zhang et al., 2021). All datasets do not contain any personally identifiable information (Zhang et al., 2021).

Originally, each dataset include three files "train.json", "valid.json" and "test.json". Following the suggestion in a reported issue of the wrench benchmark (Zhang, 2022b), we combine all three files to get a single matrix $X$ and single ground-truth label vector $y$ for the experiments on label aggregation. We then split the datasets using the original split for the experiment on end model (Appendix F.2). The information of the LFs as well as the raw data for each dataset can be found in the wrench benchmark project on Github (Zhang, 2022a).

**Baselines.** For each baseline, we use existing open-source implementations. The implementations of DS, DP, FS, MeTaL, EBCC, NPLM, and ACML-CC are from (sukrutrao, 2022), (snorkel team, 2022a), (HazyResearch, 2022), (snorkel team, 2022b), (yuan li, 2022), (BatsResearch, 2022a),

and (BatsResearch, 2022b) respectively. For baselines that require class weights as priors, we report the best results from using uniform weights and using the weights estimated by majority vote.

**Setup in Semi-supervised Label Aggregation.** When sampling $N_{gt}$ data points as the data points with known labels, we make sure that each class has at least two data points. For random forest, we use the scikit-learn implementation (rfs, 2022). When training the random forest classifier, we use five fold cross validation to perform grid search for the "max_depth" parameter in range [2, 4, 8, 16, 32, None] and the "min_samples_split" parameter in range [2, 5]. The AMCL-CC method does not support abstention, to make it work we fill in the abstentions with labels provided by majority vote; AMCL-CC requires a lot of memory on some datasets and involves solving a constrained linear programming problem which may not have a solution. When AMCL-CC fails due to out-of-memory error or no-solution-found error, we use the results from random forest. We repeat five runs and report results with error bars in Figure 2.

**Setup in Ablation Study.** For model architecture, we test two baselines. The first one is based on MLP. The input is a flattened vector of a fixed size matrix $2000 \times 50$ (padded with zero if the input matrix is smaller) and the network has 10 linear layers. The second one is based on DeepSet (Zaheer et al., 2017) where each row of $X$ is treated as a set. We use an open source implementation (manzilzaheer, 2022). We replace each of our GNN layer with a DeepSet layer.

**Setup in End Model Experiment.** When training the end model, the training set and validation set both use generated labels by each method and the test set uses ground-truth labels. For the two end model MLP and BERT, we use the implementation provided by the benchmark (Zhang, 2022a; Zhang et al., 2021). We use grid search to tune hyper-parameters for each end model based on validation set performance. We use the same search space as the benchmark (Zhang et al., 2021), as summarized in Table 7. We repeat five runs and report the scores averaged over runs in Table 6.

Table 7: Hyper-parameters and search space for the end models.

| End Model | Hyper-parameter | Description | Range |
|---|---|---|---|
| MLP | batch_size | batch size | 32,128,512 |
| | lr | learning rate | 1e-5,1e-4,1e-3,1e-2,1e-1 |
| | weight_decay | weight decay | 1e-5,1e-4,1e-3,1e-2,1e-1 |
| | ffn_num_layer | number of MLP layers | 2 |
| | ffn_hidden_size | hidden size of MLP layers | 100 |
| BERT | batch_size | batch size | 16,32 |
| | lr | learning rate | 2e-5,3e-5,5e-5 |

