# OpenReview forum: "Learning Hyper Label Model for Programmatic Weak Supervision"
_ICLR.cc/2023/Conference — ICLR 2023 poster_

### Official Review · Reviewer_YMpc · 2022-10-22

**Confidence:** 5
**Correctness:** 3
**Technical Novelty And Significance:** 3
**Empirical Novelty And Significance:** 3
**Recommendation:** 6

**Clarity, Quality, Novelty And Reproducibility:**

Clarity:
The paper is well-written and clear to follow.

Quality:
I have a few questions about the method and results.

1. How do the theoretical results adjust for class-imbalanced data? This is the common case in practice, so understanding the role of class balance in the theoretical results is important.
2. Is the "weaker form" of the better-than-random assumption actually weaker for imbalanced labeling functions?
3. Questions about the efficiency results: Can you report training time of all the methods? Can you report runtime on CPU as well? Some results, such as METAL's low running time do not match practical experience -- since METAL requires training a new label model for each dataset.

Novelty:
This problem statement is novel to the best of my knowledge. Parts of the approach have analogues in meta-learning/pretraining.

Reproducibility:
Paper seems reproducible from the writing.

**Strength And Weaknesses:**

+ Clear idea, clear evaluation.
+ The method seems to be empirically well-suppported in terms of accuracy of the final model
+ Theoretical support for existence of an optimal label model and conditions for finding one

- Theoretical analysis is unclear in the case of class-imbalanced data (which is the common case in practice)
- Experiments on efficiency a little incomplete (see Q's below)

**Summary Of The Paper:**

This paper proposes a "hyper label model" for weak supervision that can infer ground-truth labels for any weak supervision dataset without retraining. The authors evaluate their hyper label model in accuracy and efficiency on 14 benchmark datasets. The hyper label model is more efficient and on average slightly more accurate.

**Summary Of The Review:**

Overall I think this paper is strong, with a clear contribution and results. I have a few questions about the results and experiments, but they can be addressed fairly easily. I plan to give this paper a score of "accept" - contingent on having my questions answered in rebuttal.

Update after the rebuttal: I am decreasing my score to a 5 because of concerns about the characterization of previous better-than-random assumptions.

Second update after the rebuttal: Raising my score to a 6 - I think the paper is would be a positive contribution to ICLR.

---

> ### Author Response · Authors · 2022-11-09
> **Response to Reviewer YMpc**
>
> Thank you for your valuable feedback! We have responded to your questions below:
>
> > W1. How do the theoretical results adjust for class-imbalanced data? This is the common case in practice, so understanding the role of class balance in the theoretical results is important.
>
> First, we note that our theoretical results do not make any specific assumptions about the data (e.g. the data is class-balanced or not). Specifically, our analysis is built on top of the notion *valid*, i.e. $\sigma(X, y)=1$, and our analysis holds for any definition of $\sigma$. This abstraction provided by $\sigma$ offers lots of flexibility. In our definition of $\sigma$, we used a per-class condition, i.e. for each class whether the majority of LFs are better-than-random, which is invariant to the size of each class.
>
> Second, empirically our method works well on imbalanced datasets. Specifically, 7 of the 14 datasets we used are imbalanced (where F1-score is the metric, see Table 1 in the paper), for example, in the SMS dataset, the pos/neg class ratio is 0.13.
>
> > W2. Is the "weaker form" of the better-than-random assumption actually weaker for imbalanced labeling functions?
>
> To clarify, the original stronger assumption is that EACH LF is better than random in EACH class. The “weaker form” is weaker in the sense that it replaces “each LF” to be “the majority of LFs”. Therefore, regardless of whether the data is imbalanced or not, the weaker form is weaker. We rephrased in Section 3 to make it clear that the original assumption is for EACH LF and in EACH class.
>
> > W3. Questions about the efficiency results: Can you report training time of all the methods? Can you report runtime on CPU as well? Some results, such as METAL's low running time do not match practical experience -- since METAL requires training a new label model for each dataset.
>
> The reported running time already includes both the training/inference time for all methods on each dataset. We added running time on CPU in Appendix F.1 in Table 5. METAL is efficient on GPU, but can require up to 18 seconds with CPU. We note that the benchmark datasets are relatively small, so they might not match the practical experience with larger datasets. Also, to avoid confusion, METAL is only included in the latest versions of snorkel and previously the default label model was not METAL but a much slower one.

---

> > ### Comment · Reviewer_YMpc · 2022-11-17
> > **Response**
> >
> > > To clarify, the original stronger assumption is that EACH LF is better than random in EACH class
> >
> > I don't believe that this is the standard weak supervision assumption. The standard assumption is the **on average**, the accuracies are better than random.
> >
> > Consider the following two scenarios. Assume everything is class-balanced, with binary labels, and all the labeling functions  are uncorrelated:
> > * **Scenario 1**: You have 5 labeling functions, 3 of them (a, b, c) have 51% accuracy, and 2 of them (d, e) have 49% accuracy.
> > * **Scenario 2**: You have 5 labeling functions, 3 of them (a, b, c) have 49% accuracy, and 2 of them (d, e) have 51% accuracy.
> >
> > In the weak supervision setting, these two scenarios are indistinguishable. But if you assume that the labeling functions are better than random on average, you are only left with scenario 1.
> >
> > The METAL paper [1] has a discussion of this ("Checking for Identifiability" in Section 4.1).
> >
> > Can you explain how your read of the weaker assumption fits in with this prior work? The paper is still has good contributions even without it, but it should be scoped to have the proper claims.
> >
> > EDIT: I will be decreasing my score to a 5 until this point is addressed, since the rebuttal update now includes some incorrect statements.
> >
> > [1] Ratner et al. Training Complex Models with Multi-Task Weak Supervision. AAAI 2019.

---

> > > ### Author Response · Authors · 2022-11-17
> > > **Re:Response**
> > >
> > > Thank you for pointing this out. Yes, we double-checked the assumptions made in existing work, and it is inaccurate to state that “the assumption is that EACH LF is better than random in EACH class” is the standard assumption. We also found that different existing methods make different assumptions[1, 2, 3, 4]. We have revised Section 2 and Section 3 and removed improper claims.
> > >
> > > As to [1], their “on average non-adversarial” assumption is quite weak, but is only for identifiability of the correct label permutation instead of the performance, the true assumption for good performance in [1] is the underlying generative distribution of the dataset follows the same form of generative model they proposed, which is different from our assumption.
> > >
> > > [1]Ratner et al. Training Complex Models with Multi-Task Weak Supervision. AAAI 2019.
> > >
> > > [2]Ratner, Alexander J., et al. "Data programming: Creating large training sets, quickly." Advances in neural information processing systems 29 (2016).
> > >
> > > [3]Fu, Daniel, et al. "Fast and three-rious: Speeding up weak supervision with triplet methods." International Conference on Machine Learning. PMLR, 2020.
> > >
> > > [4]Arachie, Chidubem, and Bert Huang. "Constrained labeling for weakly supervised learning." Uncertainty in Artificial Intelligence. PMLR, 2021.

---

> > > > ### Comment · Reviewer_YMpc · 2022-11-21
> > > > **Response**
> > > >
> > > > Thank you for addressing my concerns. You are right in that the statements of the "non-adversarial" assumption in prior work differ in their exact statements, though I believe they're all similar in spirit to "better than random on average" or something to that effect.
> > > >
> > > > You're also right that the previous works make an assumption that the LF's match the underlying generative distribution - though there have been some attempts to address this, e.g. [1].
> > > >
> > > > I think the paper does still have a good contribution to make, though it does not break (as much) new ground on the theoretical front. I'm satisfied with the edits for correctness, so I am raising my score to a 6, to reflect that the paper would be a positive contribution to ICLR.
> > > >
> > > > [1] Chen et al. Comparing the Value of Labeled and Unlabeled Data in Method-of-Moments Latent Variable Estimation. AISTATS 2021.

---

> > > > > ### Author Response · Authors · 2022-11-21
> > > > > **Response**
> > > > >
> > > > > Thank you for your reply! And thank you again for your comments that helped us to scope our work more properly.

---

> > > > > ### Author Response · Authors · 2022-11-22
> > > > > **Response**
> > > > >
> > > > > Thanks again for your reply.
> > > > > To further clarify, indeed there are existing SEMI-supervised methods[1, 2] to address the assumption that LFs match the underlying generative distribution by utilizing some labeled data. However, we note that, to the best of our knowledge, our method is the first unsupervised method that does not rely on the generative distribution assumption.
> > > > >
> > > > > As to the theoretical front, we would like to highlight that our work provides the first analytical form for label aggregation, and is the first method that does not require ad-hoc training for each dataset.
> > > > >
> > > > > [1] Chen et al. Comparing the Value of Labeled and Unlabeled Data in Method-of-Moments Latent Variable Estimation. AISTATS 2021.
> > > > >
> > > > > [2] Mazzetto, Alessio, et al. "Adversarial multi class learning under weak supervision with performance guarantees." International Conference on Machine Learning. PMLR, 2021.

---

### Official Review · Reviewer_jYPK · 2022-10-24

**Confidence:** 3
**Correctness:** 2
**Technical Novelty And Significance:** 3
**Empirical Novelty And Significance:** 3
**Recommendation:** 6

**Clarity, Quality, Novelty And Reproducibility:**

The paper is easy to read and understand.The general problem tackled in this paper (reducing annotation cost) is interesting. The proposed approach seems novel and shows better results than baselines. The code is provided as supplementary material so it should not be too difficult to reproduce the results.

**Strength And Weaknesses:**

**Strengths**
-The general problem tackled in this paper is interesting. Reducing annotation cost is an interesting research direction because it is cheaper to create new datasets or extend existing datasets.
- I like the proposed assumption “the majority of labeling functions are better than random” because it seems to be more realistic as claimed in the paper.
- I like the assumptions about the model architecture: (1) ability to accept arbitrary input size; (2) invariance to permutation of labeling functions. (3) equivariance to permutation of data points
The paper presented well the motivations behind these assumptions and how the proposed GNN model respects these assumptions.
- I like the idea of training the hyper label model on synthetic data to not increase the annotation cost. It is also possible to improve the hyper label model performances by generating more data. I also like the fact that the hyper label model is trained only one time and can be used on multiple datasets without retraining.
- The hyper label model can work with missing weak labels.
- The proposed approach shows better performances on average on multiple datasets (Table 2). The evaluation is performed in both unsupervised and semi-supervised settings. The proposed method seems faster than most of the baselines. An ablation study shows the impact of each key contributions


**Weaknesses**
- I think some parts of Section 4 are a bit misleading because p(y|X) is approximated by a uniform distribution, so theorem 1 is an optimal solution of an approximate problem. There is no strong evidence the uniform distribution is the best approximation for p(y|X), and should maybe be presented as a good approximation instead of the best. I agree that the uniform distribution has optimalities in both the worst case and the average case, but it does not mean it is always optimal. *[post rebuttal] The paper has been updated to take into account this comment and remove the misleading part.*
- I am not sure I understand the motivation of using a GNN instead of a simpler set-based model like DeepSet [A]. I agree that a GNN respects the proposed assumptions but I am not sure to understand why modeling the edges is important. I think adding an experiment with a DeepSet could help to justify this design choice. *[post rebuttal] The rebuttal gives a motivation to this design choice.*
- The paper claims the data generation process is efficient but the probability of generating a valid pair is about 0.2, which does not seem very efficient. Maybe it could be interesting to propose a better generation process or to not claim the data generation process is efficient. *[post rebuttal] The paper has been updated to remove the claim.*
- The results about the training of the hyper label model are in the supplementary and not in the main paper. I think the main paper should show at least some of them because it seems to be a key contribution of the paper. I would suggest reducing Section 3 and 4, by moving some parts to the supplementary, to have space to present some results about the training of the hyper label model.
- The related section does not mention some other approaches to reduce the annotation cost like self-supervised learning methods. I think the related work section should discuss some of the other methods.  *[post rebuttal] The related work section has been updated to cover other approaches.*

[A] Zaheer M., Kottur S., Ravanbakhsh S., Poczos B., Salakhutdinov R., Smola A. Deep Sets. In NeurIPS, 2017.


**Summary Of The Paper:**

This paper introduces a new approach that uses programmatic weak supervision. First, labeling functions are used to generate weak labels for each example. Then, a hyper label model is used to infer the ground-truth labels from the weak labels. The hyper label model is Graph Neural Network (GNN) that is trained on synthetic data, and it can be used on any new dataset without retraining. The paper also introduces a weaker assumption about the labeling functions:  the majority of labeling functions have to be better than random. The proposed approach is evaluated on 14 classification datasets and shows better results on average.

**Summary Of The Review:**

Overall, I think the paper is good and interesting but I have some concerns about one of the contributions and some design choices. Please check the Strength And Weaknesses section for more information.

---

> ### Author Response · Authors · 2022-11-09
> **Response to Reviewer jYPK**
>
> We thank you for your valuable feedback and suggestion for a new baseline. We are very happy to hear that you found the paper interesting!
>
> > W1. theorem 1 is an optimal solution of an approximate problem. uniform distribution should maybe be presented as a good approximation instead of the best.
>
> We agree Section 4 can be a bit misleading. We rephrased the text to reflect that the uniform distribution is a good (instead of the best) approximation for $p(\boldsymbol{y}|X)$. We also rephrased Theorem 1 to reflect that $h^*$ is an optimal solution of an approximate problem.
>
> > W2. I am not sure I understand the motivation of using a GNN instead of a simpler set-based model like DeepSet [A]. I am not sure to understand why modeling the edges is important. I think adding an experiment with a DeepSet could help to justify this design choice.
>
> We agree adding an experiment with DeepSet could help to justify our design choice. We have tested with DeepSet and updated our Section 6.3 (Ablation Study) and Table 4 accordingly. DeepSet has an averaged score of 61.8 which is much less than the score of LELA, 69.0.
>
> We also provide our reasoning on why DeepSet does not work well in our case:
> DeepSet models each row of $X$ as a set and losses the information of whether weak labels across data points are from the same LF or not. For example,  $X=${{1, 0, 1}, {0, 1, 0}}, with DeepSet, the information that weak labels X[0,1] and X[1, 1] are from the same LF is lost. In contrast, with a GNN, we are able to capture that information by connecting weak labels (nodes in GNN) from the same LF with solid yellow edges (see Figure 1 in paper).
>
> > W3. The paper claims the data generation process is efficient but the probability of generating a valid pair is about 0.2, which does not seem very efficient. Maybe it could be interesting to propose a better generation process or to not claim the data generation process is efficient.
>
> We rephrased the text and removed the claim that data generation is efficient.
>
> > W4. The results about the training of the hyper label model are in the supplementary and not in the main paper. I think the main paper should show at least some of them because it seems to be a key contribution of the paper.
>
> Yes, training the hyper label model is also important. We added an overview of the training process in the experiment section in the Implementation paragraph.
>
> > W5. The related section does not mention some other approaches to reduce the annotation cost like self-supervised learning methods.
>
> We added a discussion in the Related Work section about other methods to reduce annotation cost including self-supervised learning and active learning.

---

### Official Review · Reviewer_AEYU · 2022-10-24

**Confidence:** 4
**Correctness:** 4
**Technical Novelty And Significance:** 2
**Empirical Novelty And Significance:** 3
**Recommendation:** 6

**Clarity, Quality, Novelty And Reproducibility:**

The paper is clearly written and the contributions are novel. The authors have shared the code and experiments are on public benchmark datasets so should be reproducible.

**Strength And Weaknesses:**

Strengths:
1. I like the idea to have a single label model that can work across various datasets and the use of GNNs as label model so that the model is invariant to the permutations of LFs.
2. The empirical evaluation on benchmark datasets is comprehensive and shows the benefit of their method.
3. They give a method to generate training data to train the hyper label model so that it can work across various datasets.

Weaknesses/Questions:
1. The assumptions in Section 3 are reasonable. However I am not quite sure if the assumption per class is a weaker form or stronger form ? It seems stronger to me? Moreover won't majority vote alone give accurate inference under such assumption? Could you please share some analysis on this aspect and justify why would one need a hyper label model when this assumption is satisfied?

2. What happens to the noise rates of different labeling functions? How are they getting accounted in the inference procedure?

3. A single solution for all datasets might be too much to ask for and it is very much possible that by optimizing for generality one looses the specific advantages while optimizing for a specific setting. Are there ways to optimize the process for the dataset at hand?


**Summary Of The Paper:**

The paper proposes a general method (hyper label model) to infer pseudo-labels in weak supervision that doesn't require learning parameters for each dataset separately. They characterize an optimal analytical model that is computationally intractable but gives true labels for any dataset under mild assumptions on labeling function -- better than random for each class. They give a novel hyper label model based on Graph neural networks to approximate this optimal analytical model. The choice of GNN is motivated by the setting -- it requires the model to be invariant to the permutation of LFs (or data points). They empirically evaluated the proposed solution on 14 real-world benchmark datasets for this problem and found that the proposed hyper label model works better than the baselines on most of the  datasets.


**Summary Of The Review:**

The paper brings in a novel and valuable perspective on learning label models for weak supervision. The evaluation is satisfactory. I have a few questions and concerns listed above, except those I liked reading the paper.

---

> ### Author Response · Authors · 2022-11-09
> **Response to Reviewer AEYU**
>
> Thank you for the helpful comments! We are very happy to hear that you liked reading the paper! We have responded to your questions below.
>
> > W1.1. The assumptions in Section 3 are reasonable. However I am not quite sure if the assumption per class is a weaker form or stronger form ? It seems stronger to me?
>
> The original stronger assumption is that EACH LF is better-than-random in EACH class. For example, we have 10 LFs: $(LF_1, ..., LF_{10})$ and two classes $(C_1, C_2)$. Let $V(LF_i, C_j)$ be a boolean value denoting whether $LF_i$ is better than random on class $C_j$. The original assumption would require all the 20 values $[V(LF_1, C_1), V(LF_1, C_2), ..., V(LF_{10}, C_1), V(LF_{10}, C_2)]$ to be True.
> Our assumption is that for each class the Majority of LFs are better than random. In our assumption, we require only 6 values from $[V(LF_1, C_1), V(LF_2, C_1), ..., V(LF_{10}, C_1)]$ to be True and 6 values from $[V(LF_1, C_2), V(LF_2, C_2), ..., V(LF_{10}, C_2)]$ to be True, totaling 12 values to be True, which is much less than the original assumption. We rephrased some text in Section 3 to make it clear that the original assumption is for EACH LF and in EACH class.
>
> >W1.2. Moreover won't majority vote alone give accurate inference under such assumption? Could you please share some analysis on this aspect and justify why would one need a hyper label model when this assumption is satisfied?
>
> Under such assumption, Majority vote is reasonably good but it is not optimal, while our proposed analytical method has optimal properties (Theorem 1). However, the analytical method is computationally intractable, so we train a hyper label model to approximate it. One can just use majority vote if one only needs reasonable inference performance, but if one wants more accurate inference, one would need the hyper label model.
>
> >W2. What happens to the noise rates of different labeling functions? How are they getting accounted in the inference procedure?
>
> Different from all existing methods, one benefit of our method is that we do not have to explicitly model the noise rates of different labeling functions and the noise is handled implicitly during inference. Specifically, the process of forward pass can be regarded as a denoising process, which is learned from our generated training data. This is in the same spirit as learned denoising in the image domain[1].
>
> [1] Liu, Jiaming, et al. "Learning raw image denoising with bayer pattern unification and bayer preserving augmentation." Proceedings of the IEEE/CVF Conference on Computer Vision and Pattern Recognition Workshops. 2019.
>
>  >W3.  Are there ways to optimize the process for the dataset at hand?
>
> We agree that optimizing the process for the dataset at hand might be able to further boost performance and we believe it is a very interesting future direction to explore. Currently, our method can only optimize for a specific dataset at hand in the form of fine-tuning using some ground-truth examples. One possible way to further optimize for a given dataset $X$ is to do self-supervised finetuning. For example, given a dataset $X$, we could generate many instances of valid $\boldsymbol{y}$, and then do finetuning. Another interesting direction to explore is to incorporate additional information about LFs e.g. dependency structure. This would require encoding the structure embeddings and then using the embeddings as extra input to our model.

---

### Official Review · Reviewer_SvzH · 2022-10-25

**Confidence:** 4
**Correctness:** 4
**Technical Novelty And Significance:** 4
**Empirical Novelty And Significance:** 4
**Recommendation:** 8

**Clarity, Quality, Novelty And Reproducibility:**

The paper proposes a brand novel model for PWS that is theoretically founded, and empirically validated in benchmark. The contribution is significant (See strengths above). The paper is well written and very pleasant to read.

Not sure if the authors will release their pretrained models, but it would definitely be very impactful to the community if they do.

**Strength And Weaknesses:**

Strengths:

1. The paper takes on a novel perspective for PWS, that trains a hyper label model that can be used off the shelf universally for any task, any list of LFs and any training data. The training is entirely unsupervised, theoretically founded (Sec 4) and tractably approximated by a data generation process (Sec 5).
2. The model can either be used on its own (as a hyper label model), or further fine tuned if some ground truth labels are used. To me the idea is an interesting counterpart in PWS to the well studied concept of "unsupervised pretraining" in Vision and NLP. IMHO this is a very refreshing first step that can inspire a whole new branch of ideas and solutions in PWS. (Perhaps the authors can also comment their view on this, since my impression is that the paper wasn't positioned this way, e.g. no mention of related works in transfer learning?)

Weaknesses: I think the paper is in a very good shape and only have some minor questions/comments/clarification.

1. There seems to be a gap between the loss functions in the theoretical analysis (L2 loss in Eq. 4) and the one used in training GNN (CE in Eq. 5). Can the authors please comment on this? In particular, I wonder what happens if L2 loss is used for training (ablation?)
2. Following the last question, have the authors considered/tried, in Step 2 (Sec 5.1), generating one X and several valid y's, and add (X, \bar{y}) as a training point (using L2 loss as now \bar{y} would not be binary anymore), and why not do this?
3. The data generation process uses a uniform distribution throughout. Have the authors tried other distributions? In particular, in Step 2 (Sec 5.1), the method simply samples each cell in X i.i.d. uniformly from {-1, 1, 0} (if I wasn't mistaken?) Real world LFs have varying degree of accuracy and correlation (Snorkel's assumption), the data generation model seems oversimplified and can be very different from real data. For instance, would it make sense to pull a large set of real world X's (e.g. combining all X's from WRENCH datasets) and only simulate y?
4. Given the difference between the synthetic vs real world distribution of X, I'm quite surprised that fine tuning didn't really improve much over the pretrained model (Fig 2). Do the authors have some intuitions?
5. Why does one have to convert to one-vs-rest binary tasks in order to train multiclass hyper label model? Perhaps I miss the point, I didn't see any blocker from simply simulating multiclass y and using multiclass CE in Eq 5?
6. The proposed model is referred to as hyper label model before Sec 6 and as LELA after, it would be good to unify them?

**Summary Of The Paper:**

The authors proposed for the first time a theoretically motivated label model for programatic weak supervision (PWS), called a hyper label model based on GNN. The model can be used to aggregate any list of labeling functions (LFs) for any task and infer ground-truth labels in a single pass, without need any dataset-specific training of the label model. Empirical results are quite strong compared to SOTA.

**Summary Of The Review:**

To the best of my knowledge, the paper is a significant contribution in PWS with a brand new idea of (pre)training a one-fits-all label model. Many readers should find the paper interesting.

---

> ### Author Response · Authors · 2022-11-09
> **Response to Reviewer SvzH**
>
> Thank you for the positive and insightful comments!
> > S2. the idea is a counterpart in PWS to "unsupervised pretraining" in Vision and NLP.
>
> Yes, we agree that our work overall resembles “unsupervised pretraining” in Vision and NLP. One distinction is that in Vision and NLP unsupervised pretraining is done on real data, while in our case pre-training is done on synthetic data. As you also pointed out in W3, we believe an interesting direction is to pre-train our model on real-world data (real-world $X$, or $(X, \boldsymbol{y})$) which incorporates prior information on what a human-developed LF looks like and may produce better results.
>
> >W1. gap between the loss functions in the theoretical analysis and in training GNN.
>
> Great observation! From a theoretical perspective, using either CE loss or L2 loss ensures the model learns to be $h^*$ (Currently Theorem 2 uses a CE loss but one can easily show Theorem 2 also holds for L2 loss). However, CE loss works better in practice as evidenced by the fact that CE loss is commonly used for classification tasks. One explanation is that the log function (in CE loss) is steeper than the squared function (in L2 loss) so bad predictions get penalized more, leading to faster convergence.
>
> >W2. generating one X and several valid y's, and add (X, \bar{y}) as a training point.
>
> This is an interesting idea! Averaging multiple valid $\boldsymbol{y}$ may reduce noise in the labels of training data, which might be helpful for model training. In our case, we implemented our data generation method motivated by Theorem 2 which doesn’t require reducing the noise of each training data point. On the other hand, we also have some intuition on why this might not be needed in our case. It has been observed that when the dataset size is large, neural networks are able to accommodate the noise[1]. In our case, we have infinite amount of training data so the inaccuracies of each single label might not affect the final trained model much. Nevertheless, this is an interesting idea to try out in future work.
>
> [1] Rolnick, David, et al. "Deep learning is robust to massive label noise." arXiv preprint arXiv:1705.10694 (2017).
>
> >W3. The data generation process uses a uniform distribution throughout. would it make sense to pull a large set of real world X's (e.g. combining all X's from WRENCH datasets) and only simulate y?
>
> Again, we designed our data generation method motivated by Theorem 2 which requires $p(\boldsymbol{y}|X)$ to be uniform but doesn’t require a specific distribution of $X$, so we just used a uniform distribution for $X$.
> Yes, it makes sense to use real-world $X$. As we discussed in our response to S2, it would be an interesting future direction to use real-world $X$ (and even $\boldsymbol{y}$) to pretrain our method. One challenge would be that there are few available real-world datasets (14 in wrench) and we might need a way to generate additional datasets based on the few real-world datasets.
>
> >W4. fine tuning didn't really improve much over the pretrained model
>
> The performance of label aggregation is upper-bounded by the quality of LFs. The upper-bound performance can be empirically measured by a random forest classifier trained on enough training examples. As shown in Figure 2, random forest converges to about 0.71, which is only about 2% higher than our method. This suggests our method might be pretty close to the upper bond so that further fine-tuning does not have a significant performance boost.
>
> > W5. Why does one have to convert to one-vs-rest binary tasks
>
> We agree there is no blocker in theory to extend our method to multi-class tasks. We have actually attempted that, but we were blocked by GPU memory limitation. Specifically, for the same number of data points and number of LFs, a data point with $K$ classes requires $K$ times more memory. Given that our simple fix works well already, we did not pursue that direction, but it would be an interesting follow-up to try on more powerful hardware or to improve memory efficiency during training.
>
> > W6. The proposed model is referred to as hyper label model before Sec 6 and as LELA after, it would be good to unify them?
>
> We used hyper label model before Sec 6 to highlight the general idea of learning a one-fits-all hyper model. In Sec 6, we refer to our learned model as LELA because our model is one specific instantiation of hyper label model. We rephrased the sentence where LELA is first introduced to make it clear that LELA is our learned hyper label model.
>
> > Not sure if the authors will release their pretrained models
>
> Our pretrained model and code are available in supplementary material.

---

### Author Response · Authors · 2022-11-09
**Thank all the reviewers!**

We thank all reviewers for your time and valuable feedback! We have provided responses to each reviewer's comments. Based on the comments, we have updated the paper (changes highlighted in blue).

Please let us know if you have additional questions and we are happy to discuss them. Thanks again!

---

### Decision · Program_Chairs · 2023-01-20

**Decision:**

Accept: poster

**Justification For Why Not Higher Score:**

During the discussion there remained some doubts about the depth of the theoretical contributions and their overall validity in practice.

**Justification For Why Not Lower Score:**

The paper has above threshold scores from all reviewers.

**Metareview: Summary, Strengths And Weaknesses:**

The papers discusses an approach to programmatic weak supervision using graph neural nets. The paper justifies the method theoretically and the empirical results are supportive. The paper is reasonably well written. In general the reviewers were in agreement that the paper makes a useful and justifiable contribution to the literature.


**Note From Pc:**

if the above contains the word "oral" or "spotlight" please see: "oral" presentation means -> notable-top-5% and "spotlight" means -> notable-top-25%. As stated in our emails, we are disassociating presentation type from AC recommendations